# LEARNING PATTERN-SPECIFIC EXPERTS FOR TIME SERIES FORECASTING UNDER PATCH-LEVEL DISTRIBUTION SHIFT

## ABSTRACT

Time series forecasting, which aims to predict future values based on historical data, has garnered significant attention due to its broad range of applications. However, real-world time series often exhibit complex non-uniform distribution with varying patterns across segments, such as season, operating condition, or semantic meaning, making accurate forecasting challenging. Existing approaches, which typically train a single model to capture all these diverse patterns, often struggle with the pattern drifts between patches and may lead to poor generalization. To address these challenges, we propose **TFPS**, a novel architecture that leverages pattern-specific experts for more accurate and adaptable time series forecasting. TFPS employs a dual-domain encoder to capture both time-domain and frequency-domain features, enabling a more comprehensive understanding of temporal dynamics. It then uses subspace clustering to dynamically identify distinct patterns across data patches. Finally, pattern-specific experts model these unique patterns, delivering tailored predictions for each patch. By explicitly learning and adapting to evolving patterns, TFPS achieves significantly improved forecasting accuracy. Extensive experiments on real-world datasets demonstrate that TFPS outperforms state-of-the-art methods, particularly in long-term forecasting, through its dynamic and pattern-aware learning approach. The data and codes are available: `https://anonymous.4open.science/r/TFPS-D001`.

## 1 INTRODUCTION

Time series forecasting plays a critical role in various domains, such as finance (Huang et al., 2024), weather (Bi et al., 2023; Wu et al., 2023b; Lam et al., 2023), traffic (Long et al., 2024; Kong et al., 2024), and others (Wang et al., 2023; Liu et al., 2023a), by modeling the relationship between historical data and future outcomes. However, the inherent complexity of time series data, including temporal dependencies and non-stationarity, poses significant challenges in achieving reliable forecasting results.

Recent works have shown the effectiveness of Transformer-based models for time series forecasting, due to their ability to capture long-range dependencies (Zhou et al., 2021; Wu et al., 2021; Liu et al., 2024a). In particular, models like PatchTST (Nie et al., 2023) operate by splitting the continuous time series into discrete patches and processing them with Transformer blocks. However, closely examining these patches shows that they often exhibit distribution shifts, which can be attributed to various factors such as concept drift (Lu et al., 2018). These shifts can manifest as sudden or gradual changes in the underlying patterns and distributions of the time series data. For example, patches corresponding to different regimes, seasons, or operating modes may have distinct statistical properties. Despite the remarkable success of time series forecasting models (Nie et al., 2023; Zeng et al., 2023; Eldele et al., 2024), they adopt the Uniform Distribution Modeling (UDM) strategy, which fails to account for the drifts and discrepancies between patches. This problem may result in poor generalization, hindering the performance of time series forecasting.

To quantify these distributional shifts, we split the ETTh1 dataset into patches and examined sudden drift and gradual drift in the time and frequency domains as shown in Figure 1. Specifically, we compute the maximum mean discrepancy (MMD) between patches and display the results as

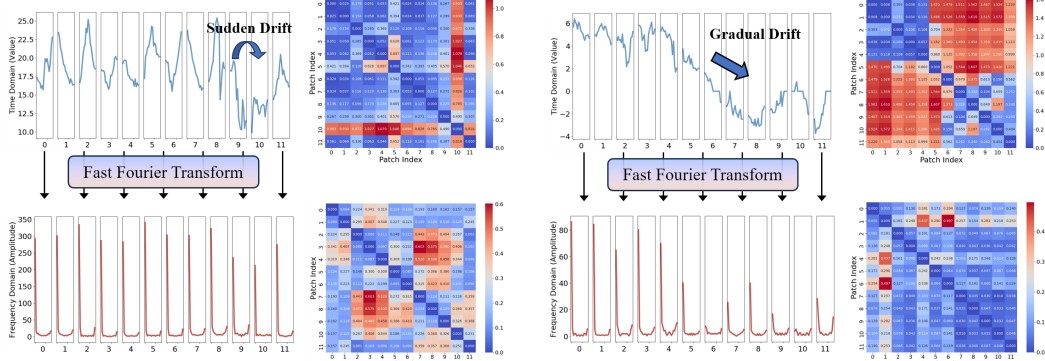

(a) Sudden drift: A new concept occurs within a short time.

(b) Gradual drift: An old concept incrementally changes to a new concept over a period of time.

Figure 1: Distribution shifts often occur between time series patches due to the non-stationarity and complex inherent in time series data. We present two examples of such shifts from the ETTh1 dataset, quantified by calculating the Maximum Mean Discrepancy (MMD) between patches. The combination of time and frequency domains offers a more comprehensive perspective of these shifts.

a heatmap. Notably, sudden drift (Figure 1 (a)) leads to patches with distinctly different distributions, while gradual drift (Figure 1 (b)) results in more pronounced drift in the time domain, making forecasting more challenging. Furthermore, the information in the frequency domain offers a complementary perspective on concept shifts. These observations highlight that time series data often exhibits a complex structure that evolves over time, with different segments having varying densities and underlying patterns (Sanakoyeu et al., 2019).

To address the challenges posed by distribution shifts in time series data, we propose a novel **T**ime-**F**requency **P**attern-**S**pecific (**TFPS**) architecture to effectively model the complex temporal patterns. In particular, TFPS consist of the following three key components. First, TFPS employs a Dual-Domain Encoder (DDE) to capture temporal dependencies in the data. DDE extracts features from both time and frequency domains to provide a comprehensive representation of the time series data, enabling the model to capture both short-term and long-term dependencies. Second, TFPS addresses the issue of concept drift by incorporating a Pattern Identifier (PI), that utilizes a subspace clustering approach to dynamically identify the distinct patterns present in different patches. Therefore, it can effectively handle nonlinear cluster boundaries and accurately assign patches to their corresponding clusters. Finally, TFPS constructs a Mixture of Pattern Experts (MoPE), a set of specialized expert models, each tailored to a specific pattern identified by the PI. By dynamically assigning patches to the appropriate expert based on their identified patterns, MoPE enables the model to effectively handle patch sequences across varying patterns and densities. This pattern-specific modeling approach allows our TFPS model to capture the unique characteristics and dynamics of each pattern, leading to improved forecasting accuracy.

In summary, the key contributions of this work are:

- We introduce a novel pattern-specific modeling strategy that decomposes the complex, evolving time series into multiple segments. Each segment is modeled by specialized experts, in contrast to the conventional UDM approach.

- We propose the TFPS framework, which explicitly addresses concept drift in time series forecasting by leveraging both time-domain and frequency-domain features. Our approach employs a clustering mechanism to dynamically assign patches to specific experts, allowing the model to better handle distributional shifts.

- We evaluate our approach on eight real-world multivariate time series datasets, demonstrating its effectiveness. Our model achieves top-1 performance in 50 out of 64 settings, showcasing its competitive edge in improving forecasting accuracy.

## 2 RELATED WORK

**Time Series Forecasting Models.** In recent years, deep models with elaborately designed architectures have achieved great progress in time series forecasting (Liu & Wang, 2024; Qiu et al., 2024). Approaches like TimesNet (Wu et al., 2023a) and ModernTCN (Luo & Wang, 2024) utilize convolutional neural networks with time-series-specific modifications, making them better suited for forecasting tasks. Additionally, simpler architectures such as Multi-Layer Perceptron (MLP)-based models (Zeng et al., 2023; Ekambaram et al., 2023) have demonstrated competitive performance. However, Transformer-based models have gained particular prominence due to their ability to model long-term dependencies in time series (Zhou et al., 2021; Wu et al., 2021; Zhou et al., 2022; Liu et al., 2024a). Notably, PatchTST (Nie et al., 2023) has become a widely adopted Transformer variant, introducing a channel-independent patching mechanism to enhance temporal representations. This approach has been further extended by subsequent models (Liu et al., 2024a; Eldele et al., 2024).

While previous work has primarily focused on capturing nonlinear dependencies in time series through enhanced model structures, our approach addresses the distribution shifts caused by evolving patterns within the data, which is a key limitation of existing methods. We emphasize tackling these shifts to improve forecasting performance in complex real-world scenarios.

**The Combination of Time and Frequency Domains.** Time-domain models excel at capturing sequential trends, while frequency-domain models are essential for identifying periodic and oscillatory patterns. Recent research has increasingly emphasized integrating information from both domains to better interpret underlying patterns. For instance, ATFN (Yang et al., 2020) demonstrates the advantage of frequency domain methods for forecasting strongly periodic time series through a time–frequency adaptive network. TFDNet (Luo et al., 2023) adopts a branching structure to capture long-term latent patterns and temporal periodicity from both domains. Similarly, JTFT (Chen et al., 2024b) utilizes the frequency domain representation to extract multi-scale dependencies while enhancing local relationships modeling through time domain representation. Yan et al. (2024) propose TFMRN, which expands data in both domains to capture finer details that may not be evident in the original data. Recently, TSLANet (Eldele et al., 2024) leverages Fourier analysis to enhance feature representation and capture both long-term and short-term interactions.

Building on these approaches, our proposed method, TFPS, introduces a novel Dual-Domain Encoder that effectively combines time and frequency domain information to capture both trend and periodic patterns. By integrating time-frequency features, TFPS significantly advances the field in addressing the complexities inherent in time series forecasting.

**Non-stationary Time Series Forecasting.** Concept drift describes unforeseeable changes in the underlying distribution of data over time, posing a significant challenge for time series forecasting (Lu et al., 2018; Fan et al., 2024). These non-stationarities complicate predictive modeling, necessitating effective solutions to handle shifting distributions. To address varying distributions, normalization techniques have emerged as a focal point in recent research, aiming to mitigate non-stationary elements and align data to a consistent distribution.

For instance, adaptive norm (Ogasawara et al., 2010) applies z-score normalization using global statistics and DAIN (Passalis et al., 2019) introduces a neural layer for adaptively normalizing each input instance. Kim et al. (2021) propose a reversible instance normalization (RevIN) to alleviate series shift. Furthermore, Non-stationary transformer (Liu et al., 2022) points that directly stationarizing time series will damage the model's capability to capture specific temporal dependencies. This work addresses the problem by introducing an innovative de-stationary attention mechanism within self-attention frameworks. Recent advancement include (Fan et al., 2023), which identifies both intra- and inter-space distribution shifts in time series data, and SAN (Liu et al., 2023b), which applies normalization at the slice level, thus opening new avenues for handling non-stationary time series data. Lastly, SIN (Han et al., 2024a) introduces a novel method to selecting the statistics and learning normalization transformations to capture local invariance in time series data.

However, over-reliance on normalization can lead to over-stationarization, where important patterns or variations in the data are smoothed out (Liu et al., 2023b). Additionally, different patterns follow distinct internal dynamics, making a unified modeling approach inefficient. Our approach brings the intrinsic non-stationarity of the original series back to latent representation, enabling better handling of distribution shifts by tailoring the experts to the evolving patterns and densities within the data.

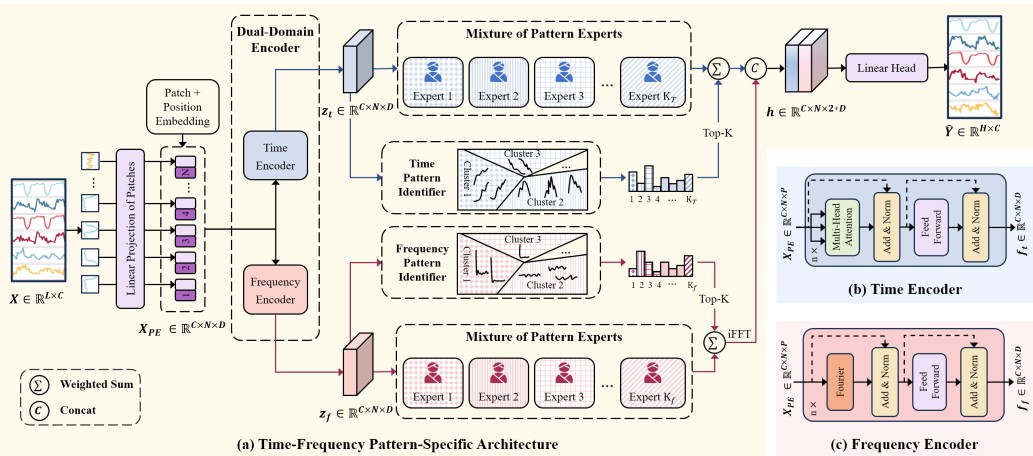

Figure 2: The structure of our proposed TFPS. The input time series is divided into patches, and positional embeddings are added. These embeddings are processed through two branches: time-domain branch and frequency-domain branch. Each branch consists of three key components: (1) an encoder to capture patch-wise features, (2) a clustering mechanism to identify patches with similar patterns, and (3) a mixture of pattern experts block to model the patterns of each cluster. Finally, the outputs from both branches are combined for the final prediction.

## 3 METHOD

### 3.1 PRELIMINARIES

Time series forecasting aims to uncover relationships between historical time series data and future data. Let $\mathcal{X}$ denote the time series, and $x_t$ represent the value at timestep $t$. Given the historical time series data $X = [x_{t-L+1}, \cdots, x_t] \in \mathbb{R}^{L \times C}$, where $L$ is the length of the look-back window and $C > 1$ is the number of features in each timestep, the objective is to predict the future series $Y = [x_{t+1}, \cdots, x_{t+H}] \in \mathbb{R}^{H \times C}$, where $H$ is the forecast horizon.

### 3.2 OVERALL ARCHITECTURE

Our model introduces three novel components: the Dual-Domain Encoder (DDE), the Pattern Identifier (PI), and the Mixture of Pattern Experts (MoPE), as illustrated in Figure 2. The DDE goes beyond traditional time-domain encoding by incorporating a frequency encoder that applies Fourier analysis, transforming time series data into the frequency domain. This enables the model to capture periodic patterns and frequency-specific features, providing a more comprehensive understanding of the data. The PI is a clustering-based module that distinguishes patches with distinct patterns, effectively addressing the variability in the data. MoPE then utilizes multiple MLP-based experts, each dedicated to modeling a specific pattern, thereby enhancing the model's ability to adapt to the temporal dynamics of time series. Collectively, these components form a cohesive framework that effectively handles concept drift between patches, leading to more accurate time series forecasting.

### 3.3 EMBEDDING LAYER

Firstly, the input sequence $X \in \mathbb{R}^{L \times C}$ is divided into patches of length $P$, resulting in $N = \lfloor \frac{(L-P)}{S} + 2 \rfloor$ tokens, where $S$ denotes the stride, defining the non-overlapping region between consecutive patches. Each patch is denoted as $\mathcal{P}_i \in \mathbb{R}^{C \times P}$. These patches are then projected into a new dimension $D$, via a linear transformation, such that, $\mathcal{P}_i \rightarrow \mathcal{P}'_i \in \mathbb{R}^{C \times D}$.

Next, positional embeddings are added to each patch to preserve the temporal ordering disrupted during the segmentation process. The position embedding for the $i$-th patch, denoted as $E_i$, is a vector of the same dimension as the projected patch. The enhanced patch is computed by summing the original patch and its positional embedding: $X_{PE_i} = \mathcal{P}'_i + E_i$, and $X_{PE} = \{X_{PE_1}, X_{PE_2}, \cdots, X_{PE_N}\}$. Notably, the positional embeddings are learnable parameters, which enables the model to capture

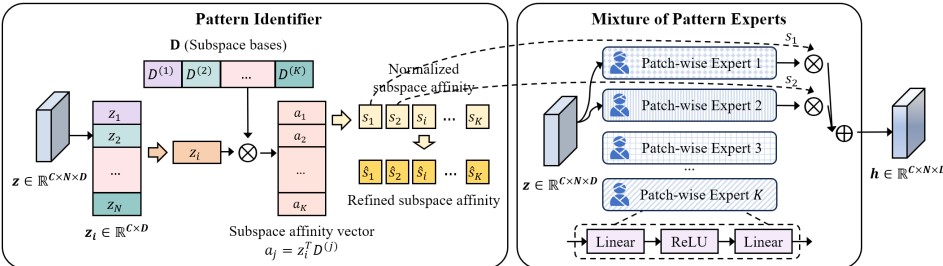

Figure 3: Illustration of the proposed Pattern Identifier and Mixture of Pattern Experts. The embedded representation $\mathbf{z}$ from DDE combines with subspace $\mathbf{D}$ to construct the subspace affinity vector, which yields the normalized subspace affinity $S$. Subsequently, the refined subspace affinity $\widetilde{S}$ is computed from $S$ to provide self-supervised information. Then, we assign the corresponding patch-wise experts to the embedded representation $\mathbf{z}$ according to $S$ for modeling.

the temporal dependencies in the time series more effectively. As a result, the final enriched patch representations are $X_{PE} \in \mathbb{R}^{C \times N \times D}$.

### 3.4 DUAL-DOMAIN ENCODER

As shown in Figure 1, both time and frequency domains reveal distinct concept drifts that can significantly affect the performance of forecasting models. To effectively address these drifts, we propose a Dual-Domain Encoder (DDE) architecture that captures both temporal and frequency dependencies inherent in time series data.

We utilize the patch-based Transformer (Nie et al., 2023) as an encoder to extract embeddings for each patch, capturing the global trend feature. The multi-head attention is employed to obtain the attention output $\mathbf{O}_t \in \mathbf{R}^{N \times D}$ as follows:

$$\mathbf{O}_t = \text{Attention}(Q, K, V) = \text{Softmax}\left(\frac{QK^T}{\sqrt{d_k}}\right)V,$$

$$Q = X_{PE}\mathbf{W}_Q, \quad K = X_{PE}\mathbf{W}_K, \quad V = X_{PE}\mathbf{W}_V. \quad (1)$$

The encoder block also incorporates BatchNorm layers and a feed-forward network with residual connections, as shown in Figure 2 (b). This process generates the temporal features $z_t \in \mathbb{R}^{C \times N \times D}$.

In parallel with the time encoder, we incorporate a Frequency Encoder by replacing the self-attention sublayer of the Transformer with a Fourier sublayer (Lee-Thorp et al., 2022). This sublayer applies a 2D Fast Fourier Transform (the number of patches, hidden dimension) to the patch representation, expressed as:

$$\mathbf{O}_f = \mathcal{F}_{patch}(\mathcal{F}_h(X_{PE})). \quad (2)$$

We only keep the real part of the result, and hence, we do not modify the feed-forward layers in the Transformer. The structure of the Frequency Encoder is depicted in Figure 2 (c), yielding frequency features $z_f \in \mathbb{R}^{C \times N \times D}$.

By modeling data in both the time and frequency domains, the DDE provides a more comprehensive understanding of the underlying patterns in time series data. This dual-domain perspective enables the model to better address complexities like concept drift and evolving temporal dynamics, ultimately improving its robustness and predictive accuracy. Through this approach, we aim to create a versatile framework capable of adapting to the intricate nature of real-world time series data.

### 3.5 PATTERN IDENTIFIER

To address the complex and evolving patterns in time series data, we introduce a novel Pattern Identifier (PI) module, an essential innovation within our framework. Unlike traditional approaches that treat the entire time series uniformly, our PI module dynamically classifies patches based on their distributional characteristics, enabling a more precise and adaptive modeling strategy.

The core of our approach lies in leveraging subspace clustering to detect concept shifts across multiple subspaces, as illustrated in Figure 3. This enables the PI module to distinguish patches by identifying their underlying patterns in both the time and frequency domains.

The PI module's uniqueness lies in its ability to directly analyze the properties of each patch, clustering them into distinct groups. In the time domain, PI allows our framework, TFPS, to detect shifts related to temporal characteristics, such as seasonality and trends, which can influence forecasting accuracy. In the frequency domain, the PI captures shifts associated with frequency-specific features, like periodic behaviors and spectral changes, offering a comprehensive view of pattern evolution across the entire time series.

This dual-domain capability represents a significant advancement in our framework, making it highly sensitive to pattern shifts occurring across both temporal and frequency dimensions. By focusing on both domains, our model overcomes the limitations of existing methods that rely solely on one representation, ensuring robustness in diverse scenarios.

To provide clarity, Figure 3 showcases an application of the PI module exclusively within the time domain. However, the insights and methodology seamlessly extend to the frequency domain, presenting a unified solution to the challenge of concept shifts.

The PI module refines subspace bases iteratively, where the improved subspaces, in turn, enhance the representation learning in deep neural networks. This iterative refinement of subspaces is a key contribution of our work, leading to more accurate identification and modeling of evolving patterns. The PI module operates in three key steps explained next.

**Construction of subspace bases.** We define a new variable $\mathbf{D} = [\mathbf{D}^{(1)}, \mathbf{D}^{(2)}, \cdots, \mathbf{D}^{(K)}]$ to represent the bases of $K$ subspaces, where $\mathbf{D}$ consists of $K$ blocks, each $\mathbf{D}^{(j)} \in \mathbf{R}^{q \times d}$, $\left\|\mathbf{D}_u^{(j)}\right\| = 1, u = 1, \cdots d, j = 1, \cdots, K$. To control the size of the columns of $\mathbf{D}$, we impose the following constraint:

$$R_1 = \frac{1}{2} \left\|\mathbf{D}^T\mathbf{D} \odot \mathbf{I} - \mathbf{I}\right\|_F^2, \tag{3}$$

where $\odot$ denotes the Hadamard product, and $\mathbf{I}$ is an identity matrix of size $Kd \times Kd$.

**Subspaces differentiation.** To ensure the dissimilarity between different subspaces, we introduce the second constraint:

$$
\begin{aligned}
R_2 &= \frac{1}{2} \left\|\mathbf{D}^{(j)T}\mathbf{D}^{(l)}\right\|_F^2, \quad j \neq l, \\
&= \frac{1}{2} \left\|\mathbf{D}^T\mathbf{D} \odot \mathbf{O}\right\|_F^2,
\end{aligned}
\tag{4}
$$

where $\mathbf{O}$ is a matrix with all off-diagonal $d$-size blocks set to 1 and diagonal blocks set to 0. Combining $D_1$ and $D_2$ yields the regularization term for $\mathbf{D}$:

$$R = \alpha(R_1 + R_2), \tag{5}$$

where $\alpha$ is a tuning parameters, fixed at $10^{-3}$ in this work.

**Subspace affinity and refinement.** We propose a novel subspace affinity measure $\mathbf{S}$ to assess the relationship between the embedded representation $\mathbf{z}$ from DDE and the subspace bases $\mathbf{D}$. The affinity $s_{ij}$, representing the probability that the embedded $\mathbf{z}_i$ belongs to the $j$-th subspace, is defined as:

$$s_{ij} = \frac{\left\|\mathbf{z}_i^T\mathbf{D}^{(j)}\right\|_F^2 + \eta d}{\sum_j(\left\|\mathbf{z}_i^T\mathbf{D}^{(j)}\right\|_F^2 + \eta d)}, \tag{6}$$

where $\eta$ is a parameter controlling the smoothness, fixed to the same value as $d$. To emphasize more confident assignments, we introduce a refined subspace affinity $\widetilde{S}$:

$$\widetilde{s}_{ij} = \frac{s_{ij}^2/\sum_i s_{ij}}{\sum_j(s_{ij}^2/\sum_i s_{ij})}. \tag{7}$$

This refinement sharpens the clustering by weighting high-confidence assignments more heavily. The subspace clustering objective based on the Kullback-Leibler divergence is then:

$$\mathcal{L}_{sub} = KL(\widetilde{S} \parallel S) = \sum_i \sum_j \widetilde{s}_{ij} log\frac{\widetilde{s}_{ij}}{s_{ij}}. \tag{8}$$

The clustering loss is defined as:

$$\mathcal{L}_{PI} = R + \beta\mathcal{L}_{sub}, \tag{9}$$

where $\beta$ is a hyperparameter balancing the regularization and subspace clustering terms.

## 3.6 MIXTURE OF PATTERN EXPERTS

Traditional time series forecasting methods often rely on a uniform distribution modeling (UDM) approach, which struggles to adapt to the complexities of diverse and evolving patterns in real-world data. To address this limitation, we introduce the Mixture of Pattern Experts module (MoPE), which assigns specialized experts to patches based on their unique underlying patterns, enabling more precise and adaptive forecasting.

Given the cluster assignments $s$ obtained from the PI module, we apply the Patch-wise MoPE to the feature tensor $z \in \mathbb{R}^{C \times N \times D}$. The MoPE module consists of the following key components:

**Gating Network:** The gating network $G$ calculates the gating weights for each expert based on the cluster assignment $s$ and selects the top $k$ experts. The gating weights are computed as:

$$G(s) = \text{Softmax}(\text{TopK}(s)). \tag{10}$$

Here, the top $k$ logits are selected and normalized using the Softmax function to produce the gating weights.

**Expert Networks:** The MoPE contains $K$ expert networks, denoted as $E_1, \ldots, E_K$. Each expert network is modeled as an MLP consisting of two linear layers and a ReLU activation. Given a patch-wise feature $z$, each expert network $E_k$ processes the input to generate its respective output.

**Output Aggregation:** The final output $h$ of the MoPE module is a weighted sum of the outputs from all the selected experts, with the weights provided by the gating network:

$$h = \sum_{k=1}^{K} G(s) E_k(z). \tag{11}$$

After the frequency branch is processed by the inverse Fast Fourier transform, the time-frequency outputs $h_t$ and $h_f$, are concatenated to form $h = \text{concat}(h_t, h_f) \in \mathbb{R}^{C \times N \times 2D}$.

Finally, a linear transformation is applied to the concatenated output $h$ to generate the prediction: $\hat{Y} = \text{Linear}(h) \in \mathbb{R}^{H \times C}$.

This approach ensures that the MoPE dynamically assigns and aggregates contributions from various experts based on the evolving patterns, improving the model's adaptability and accuracy.

## 3.7 LOSS FUNCTION

Following the approach outlined in Nie et al. (2023), we use the Mean Squared Error (MSE) loss to quantify the discrepancy between predicted values $\hat{Y}$ and ground truth values $Y$: $\mathcal{L}_{MSE} = (\hat{Y} - Y)^2$. In addition to the MSE loss, we incorporate the clustering regularization loss from the PI module, yielding the final loss function:

$$\mathcal{L} = \mathcal{L}_{MSE} + \mathcal{L}_{PI_t} + \mathcal{L}_{PI_f}. \tag{12}$$

This combined loss ensures that the model not only minimizes forecasting errors but also accurately identifies and maintains the integrity of pattern clusters across time. The algorithm is provided in the Appendix G.

## 4 EXPERIMENTS

In this section, we present experimental results to demonstrate the effectiveness of our proposed TFPS framework, including its forecasting performance and model analysis.

### 4.1 EXPERIMENTAL SETUP

**Dataset and Baselines.** We conducted our experiments on eight publicly available real-world multivariate time series datasets, i.e., ETT (ETTh1, ETTh2, ETTm1, ETTm2), Exchange, Weather, Electricity, and ILI. These datasets are provided in (Wu et al., 2021) for time series forecasting. We followed the standard protocol for data preprocessing. Specifically, we split the datasets into

Table 1: The statistics of the datasets.

| Datasets | ETTh1 & ETTh2 | ETTm1 & ETTm2 | Exchange-Rate | Weather | Electricity | ILI |
|---|---|---|---|---|---|---|
| Variates | 7 | 7 | 8 | 21 | 321 | 7 |
| Timesteps | 17,420 | 69,680 | 7,588 | 52,696 | 26,304 | 966 |
| Granularity | 1 hour | 15 min | 1 day | 10 min | 1 hour | 1 week |

training, validation, and testing by a ratio of 6:2:2 for the ETT dataset and 7:1:2 for the other dataset (Zeng et al., 2023). Table 1 shows the statistics of these datasets.

In our experiments, we employed a diverse set of state-of-the-art forecasting models as baselines, categorized based on the type of information they utilize as follows. **(1) Time-domain methods:** PatchTST (Nie et al., 2023), DLinear (Zeng et al., 2023), TimesNet (Wu et al., 2023a) and iTransformer (Liu et al., 2024a); **(2) Frequency-domain methods:** FEDformer (Zhou et al., 2022) and FITS (Xu et al., 2024); **(3) Time-frequency methods:** TFDNet-IK (Luo et al., 2023) and TSLANet (Eldele et al., 2024). We rerun all the experiments with codes provided by their official implementation.

Note that some recent foundation models such as Time-LLM (Jin et al., 2024) and MOIRAI (Woo et al., 2024) have demonstrated remarkable performance in time series forecasting by leveraging knowledge from diverse datasets through pretraining. However, our TFPS model and the baselines above focus on specific dataset for training and testing. Therefore, we did not include these pretrained foundation models in comparison.

**Experiments details.** Following previous works (Nie et al., 2023), we used ADAM (Kingma & Ba, 2014) as the default optimizer across all the experiments. We employed the MSE and mean absolute error (MAE) as the evaluation metrics, and a lower MSE/MAE value indicates a better performance. TFPS was implemented by PyTorch (Paszke et al., 2019) and trained on a single NVIDIA RTX 3090 24GB GPU. We conducted grid search to optimize the following three parameters, i.e., learning rate $= \{0.0001, 0.0005, 0.001, 0.005, 0.01, 0.05\}$, the number of experts in the time domain $K_t = \{1, 2, 4, 8\}$, and the number of experts in the frequency domain $K_f = \{1, 2, 4, 8\}$.

## 4.2 OVERALL PERFORMANCE COMPARISON

Table 2 highlights the consistent superiority of TFPS across multiple datasets and prediction horizons, securing the top performance in 50 out of 64 experimental configurations. In particular, TFPS demonstrates significant improvements over time-domain methods, with an overall improvement of 8.7% in MSE and 5.9% in MAE. Compared to frequency-domain methods, TFPS shows even more pronounced enhancements, with MSE improved by 15.8% and MAE by 11.6%.

While the time-frequency methods like TSLANet and TFDNet perform competitively on several datasets, TFPS still outperforms them, showing improvement of 4.5% in MSE and 1.9% in MAE. These substantial improvements can be attributed to the integration of both time- and frequency-domain information, combined with our innovative approach to modeling distinct patterns with specialized experts. By addressing the underlying concept shifts and capturing complex, evolving patterns in time series data, TFPS achieves more accurate predictions than other baseline models.

## 4.3 ABLATION STUDY

Table 3 presents the MSE results of TFPS and its variants with different combinations of encoders, PI, and MoPE. **1) Best Result.** The full TFPS model, i.e., both the time and frequency branches, along with their respective encoders, PI, and MoPE are included, performs the best across all the forecast horizons for both datasets. **2) Linear vs. PI.** We replace PI with a linear layer and find that it generally results in higher MSE in most cases, indicating that accurately capturing specific patterns is crucial. **3) Impact of Pattern-aware Modeling.** Additionally, when comparing the results with the encoder-only configuration, two variants with MoPE in each branch achieved improved MSE, further supporting the necessity of patter-aware modeling. **4) Importance of DDE.** Furthermore, we find that both the time encoder and frequency encoder alone yield worse performance, with the time encoder playing a more significant role. In summary, incorporating both branches with PI and MoPE provides the best performance, while simpler configurations result in higher MSE.

Table 2: Multivariate long-term forecasting results with prediction lengths $H \in \{24, 36, 48, 60\}$ for ILI and $H \in \{96, 192, 336, 720\}$ for others. The input lengths are $L = 104$ for ILI and $L = 96$ for others. The best results are highlighted in **bold** and the second best are underlined.

| Model | | IMP. | TFPS (Our) | | TSLANet (2024) | | FITS (2024) | | iTransformer (2024a) | | TFDNet-IK (2023) | | PatchTST (2023) | | TimesNet (2023a) | | DLinear (2023) | | FEDformer (2022) | |
|---|---|---|---|---|---|---|---|---|---|---|---|---|---|---|---|---|---|---|---|---|
| Metric | | MSE | MSE | MAE | MSE | MAE | MSE | MAE | MSE | MAE | MSE | MAE | MSE | MAE | MSE | MAE | MSE | MAE | MSE | MAE |
| ETTh1 96 | | -1.1% | 0.398 | 0.413 | 0.387 | 0.405 | 0.395 | **0.403** | 0.387 | 0.405 | 0.396 | 0.409 | 0.413 | 0.419 | 0.389 | 0.412 | 0.398 | 0.410 | **0.385** | 0.425 |
| ETTh1 192 | | 4.8% | **0.423** | **0.423** | 0.448 | 0.436 | 0.445 | 0.432 | 0.441 | 0.436 | 0.451 | 0.441 | 0.460 | 0.445 | 0.441 | 0.442 | 0.434 | 0.427 | 0.441 | 0.461 |
| ETTh1 336 | | 1.8% | **0.484** | **0.461** | 0.491 | 0.487 | 0.491 | 0.463 | 0.491 | 0.463 | 0.495 | 0.462 | 0.497 | 0.463 | 0.491 | 0.467 | 0.499 | 0.477 | 0.491 | 0.473 |
| ETTh1 720 | | 3.0% | **0.488** | **0.476** | 0.505 | 0.486 | 0.496 | 0.485 | 0.509 | 0.494 | 0.492 | 0.482 | 0.501 | 0.486 | 0.512 | 0.491 | 0.508 | 0.503 | 0.501 | 0.499 |
| ETTh2 96 | | -2.0% | 0.313 | 0.355 | 0.290 | 0.345 | 0.295 | 0.344 | 0.301 | 0.350 | **0.289** | **0.337** | 0.299 | 0.348 | 0.324 | 0.368 | 0.315 | 0.374 | 0.342 | 0.383 |
| ETTh2 192 | | -2.9% | 0.405 | 0.410 | **0.362** | **0.391** | 0.382 | 0.396 | 0.380 | 0.399 | 0.379 | 0.395 | 0.383 | 0.398 | 0.393 | 0.410 | 0.432 | 0.447 | 0.434 | 0.440 |
| ETTh2 336 | | 10.5% | **0.392** | **0.415** | 0.401 | 0.419 | 0.416 | 0.425 | 0.424 | 0.432 | 0.416 | 0.422 | 0.424 | 0.431 | 0.429 | 0.437 | 0.486 | 0.481 | 0.512 | 0.497 |
| ETTh2 720 | | 12.6% | **0.410** | **0.433** | 0.419 | 0.439 | 0.418 | 0.437 | 0.430 | 0.447 | 0.424 | 0.441 | 0.429 | 0.445 | 0.433 | 0.448 | 0.732 | 0.614 | 0.467 | 0.476 |
| ETTm1 96 | | 4.1% | **0.327** | **0.367** | 0.329 | 0.368 | 0.354 | 0.375 | 0.342 | 0.377 | 0.331 | 0.369 | 0.331 | 0.370 | 0.337 | 0.377 | 0.336 | 0.374 | 0.360 | 0.406 |
| ETTm1 192 | | 2.6% | **0.374** | 0.395 | 0.376 | 0.383 | 0.392 | 0.393 | 0.383 | 0.396 | 0.376 | **0.381** | 0.374 | 0.395 | 0.395 | 0.406 | 0.382 | 0.392 | 0.395 | 0.427 |
| ETTm1 336 | | 4.2% | **0.401** | **0.408** | 0.403 | 0.414 | 0.425 | 0.415 | 0.418 | 0.418 | 0.405 | 0.410 | 0.402 | 0.412 | 0.433 | 0.432 | 0.414 | 0.414 | 0.448 | 0.458 |
| ETTm1 720 | | -0.7% | **0.445** | **0.438** | 0.479 | 0.456 | 0.486 | 0.449 | 0.487 | 0.457 | 0.471 | 0.466 | 0.466 | 0.446 | 0.484 | 0.458 | 0.478 | 0.455 | 0.491 | 0.479 |
| ETTm2 96 | | 6.9% | **0.170** | **0.255** | 0.179 | 0.261 | 0.183 | 0.266 | 0.186 | 0.272 | 0.176 | 0.267 | 0.177 | 0.260 | 0.182 | 0.262 | 0.184 | 0.276 | 0.193 | 0.285 |
| ETTm2 192 | | 7.1% | **0.235** | **0.296** | 0.243 | 0.303 | 0.247 | 0.305 | 0.254 | 0.314 | 0.245 | 0.302 | 0.248 | 0.306 | 0.252 | 0.307 | 0.282 | 0.357 | 0.256 | 0.324 |
| ETTm2 336 | | 4.6% | **0.297** | **0.335** | 0.308 | 0.345 | 0.307 | 0.342 | 0.316 | 0.351 | 0.303 | 0.340 | 0.303 | 0.341 | 0.312 | 0.346 | 0.324 | 0.364 | 0.321 | 0.364 |
| ETTm2 720 | | 3.6% | **0.401** | **0.397** | 0.403 | 0.400 | 0.407 | 0.401 | 0.414 | 0.407 | 0.405 | 0.399 | 0.405 | 0.403 | 0.417 | 0.404 | 0.441 | 0.454 | 0.434 | 0.426 |
| Exchange 96 | | 12.7% | **0.083** | **0.205** | 0.085 | 0.206 | 0.088 | 0.210 | 0.086 | 0.206 | 0.084 | 0.205 | 0.089 | 0.206 | 0.105 | 0.233 | 0.089 | 0.219 | 0.136 | 0.265 |
| Exchange 192 | | 11.2% | **0.174** | **0.297** | 0.178 | 0.300 | 0.181 | 0.304 | 0.181 | 0.304 | 0.176 | 0.299 | 0.178 | 0.302 | 0.219 | 0.342 | 0.180 | 0.319 | 0.279 | 0.384 |
| Exchange 336 | | 10.4% | **0.310** | **0.398** | 0.329 | 0.415 | 0.324 | 0.413 | 0.338 | 0.422 | 0.321 | 0.409 | 0.326 | 0.411 | 0.353 | 0.433 | 0.313 | 0.423 | 0.465 | 0.504 |
| Exchange 720 | | -13.3% | 1.011 | 0.756 | 0.850 | 0.693 | 0.846 | 0.696 | 0.853 | 0.696 | **0.835** | **0.689** | 0.840 | 0.690 | 0.912 | 0.724 | 0.837 | 0.690 | 1.169 | 0.826 |
| Weather 96 | | 15.6% | **0.154** | **0.202** | 0.176 | 0.216 | 0.167 | 0.214 | 0.176 | 0.216 | 0.165 | 0.209 | 0.177 | 0.219 | 0.168 | 0.218 | 0.197 | 0.257 | 0.236 | 0.325 |
| Weather 192 | | 10.6% | **0.205** | **0.249** | 0.226 | 0.258 | 0.215 | 0.257 | 0.225 | 0.257 | 0.214 | 0.252 | 0.225 | 0.259 | 0.226 | 0.267 | 0.237 | 0.294 | 0.268 | 0.337 |
| Weather 336 | | 9.1% | **0.262** | **0.289** | 0.279 | 0.299 | 0.270 | 0.299 | 0.281 | 0.299 | 0.267 | 0.298 | 0.278 | 0.298 | 0.283 | 0.305 | 0.283 | 0.332 | 0.366 | 0.402 |
| Weather 720 | | 4.1% | **0.344** | **0.342** | 0.355 | 0.355 | 0.347 | 0.345 | 0.358 | 0.350 | 0.347 | 0.346 | 0.351 | 0.346 | 0.355 | 0.353 | 0.347 | 0.382 | 0.407 | 0.422 |
| Electricity 96 | | 14.6% | **0.149** | **0.236** | 0.155 | 0.249 | 0.200 | 0.278 | 0.151 | 0.241 | 0.171 | 0.254 | 0.166 | 0.252 | 0.168 | 0.272 | 0.195 | 0.277 | 0.189 | 0.304 |
| Electricity 192 | | 12.0% | **0.162** | **0.253** | 0.170 | 0.264 | 0.200 | 0.281 | 0.167 | 0.258 | 0.189 | 0.269 | 0.174 | 0.261 | 0.186 | 0.289 | 0.194 | 0.281 | 0.198 | 0.312 |
| Electricity 336 | | 0.2% | 0.200 | 0.310 | 0.197 | 0.282 | 0.214 | 0.295 | **0.179** | **0.271** | 0.205 | 0.284 | 0.190 | 0.277 | 0.197 | 0.298 | 0.207 | 0.296 | 0.212 | 0.326 |
| Electricity 720 | | 7.2% | **0.220** | 0.320 | 0.224 | 0.318 | 0.256 | 0.328 | 0.229 | 0.319 | 0.247 | 0.318 | 0.230 | **0.312** | 0.225 | 0.322 | 0.243 | 0.330 | 0.242 | 0.351 |
| ILI 24 | | 40.9% | **1.349** | **0.760** | 1.749 | 0.898 | 3.489 | 1.373 | 2.443 | 1.078 | 1.824 | 0.824 | 1.614 | 0.835 | 1.699 | 0.871 | 2.239 | 1.041 | 3.217 | 1.246 |
| ILI 36 | | 43.6% | **1.239** | **0.752** | 1.754 | 0.912 | 3.530 | 1.370 | 2.455 | 1.086 | 1.699 | 0.813 | 1.475 | 0.859 | 1.733 | 0.913 | 2.238 | 1.049 | 2.688 | 1.074 |
| ILI 48 | | 40.4% | **1.461** | **0.801** | 2.050 | 0.984 | 3.671 | 1.391 | 3.437 | 1.331 | 1.762 | 0.831 | 1.642 | 0.880 | 2.272 | 0.999 | 2.252 | 1.064 | 2.540 | 1.057 |
| ILI 60 | | 39.8% | **1.458** | **0.836** | 2.240 | 1.039 | 4.030 | 1.462 | 2.734 | 1.155 | 1.758 | 0.863 | 1.608 | 0.885 | 1.998 | 0.974 | 2.236 | 1.057 | 2.782 | 1.136 |
| 1$^{st}$ Count | | | 50 | | 3 | | 1 | | 2 | | 6 | | 1 | | 0 | | 0 | | 1 | |

Table 3: Ablation study of TFPS components. The model variants in our ablation study include the following configurations across both time and frequency branches: (a) inclusion of the encoder, PI and MoPE; (b) PI replaced with Linear; (c) only the encoder. The best results are in **bold**.

| Time Branch | | | Frequency Branch | | | ETTh1 | | | | ETTh2 | | | |
|---|---|---|---|---|---|---|---|---|---|---|---|---|---|
| Encoder | PI | MoPE | Encoder | PI | MoPE | 96 | 192 | 336 | 720 | 96 | 192 | 336 | 720 |
| ✓ | ✓ | ✓ | ✓ | ✓ | ✓ | **0.398** | **0.423** | **0.484** | **0.488** | **0.313** | **0.405** | **0.392** | **0.410** |
| ✓ | ✓ | ✓ | | | | 0.401 | 0.459 | 0.486 | 0.492 | 0.318 | 0.409 | 0.400 | 0.428 |
| ✓ | Linear | ✓ | | | | 0.401 | 0.451 | 0.494 | 0.509 | 0.325 | 0.411 | 0.400 | 0.434 |
| ✓ | | | | | | 0.414 | 0.460 | 0.501 | 0.500 | 0.339 | 0.411 | 0.426 | 0.431 |
| | | | ✓ | ✓ | ✓ | 0.455 | 0.507 | 0.539 | 0.576 | 0.324 | 0.407 | 0.417 | 0.436 |
| | | | ✓ | Linear | ✓ | 0.503 | 0.535 | 0.558 | 0.583 | 0.398 | 0.446 | 0.457 | 0.444 |
| | | | ✓ | | | 0.552 | 0.583 | 0.591 | 0.594 | 0.371 | 0.426 | 0.418 | 0.463 |

## 4.4 COMPARISON WITH NORMALIZATION METHODS

Normalization methods such as SIN (Han et al., 2024a), Dish-TS (Fan et al., 2023), and Non-Stationary Transformers (Liu et al., 2022) can reduce fluctuations to enhance performance and are widely used for non-stationary time series forecasting. We compare our TFPS with these state-of-the-art normalization methods and Table 10 presents the average MSE evaluation across all forecasting lengths for each dataset. While normalization methods contribute to data stabilization, TFPS provides a more nuanced approach by leveraging distribution-specific modeling, leading to significant improvements with an average MSE decrease of 24.3%. Detailed results for all cases can be found in Appendix E.2.

Table 4: Comparison between TFPS and normalization approaches.

| Model | TFPS | FEDformer | | |
|---|---|---|---|---|
| | | + SIN | + Dish-TS | + NST |
| ETTh1 | **0.448** | 0.458 | 0.461 | 0.456 |
| ETTh2 | **0.380** | 0.501 | 1.005 | 0.481 |
| ETTm1 | **0.395** | 0.409 | 0.422 | 0.411 |
| ETTm2 | **0.276** | 0.437 | 0.759 | 0.315 |
| Weather | **0.241** | 0.326 | 0.398 | 0.268 |

## 4.5 VISUALIZATION

We visualize the prediction curves for ETTh1 with $H = 192$. Given that DLinear exhibits competitive performance, we compare its results with those of TFPS in Figure 4 under two scenarios: (a) sudden drift caused by external factors or random events, and (b) gradual drift where the trend is

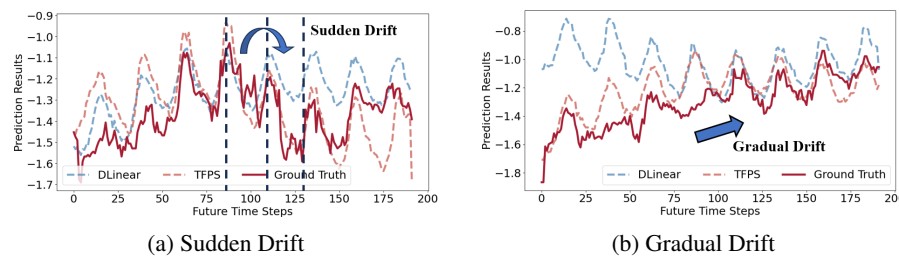

(a) Sudden Drift          (b) Gradual Drift

Figure 4: Visualizations of DLinear and TFPS on the ETTh1 dataset when $H = 192$.

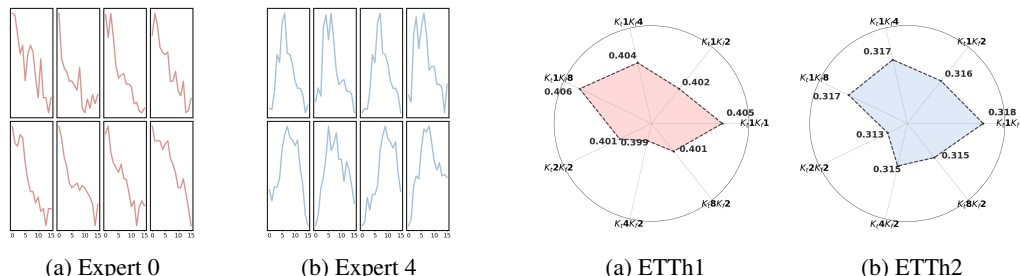

(a) Expert 0      (b) Expert 4         (a) ETTh1      (b) ETTh2

Figure 5: Interpretable patterns via PI. Expert-0 specializes in downward trends, while Expert-4 focuses on parabolic trends.

Figure 6: Experiments on the number of experts when $H = 96$. Detailed results are provided in Appendix E.1.

dominant. It is evident that DLinear struggles to achieve accurate predictions in both scenarios. In contrast, our TFPS consistently produces accurate forecasts despite these challenges, demonstrating its robustness in dealing with various concept dynamics.

### 4.6 ANALYSIS OF EXPERTS

**Qualitative Visualizations of Pattern Identifier.** Through training, pattern experts in MoPE spontaneously specialize, and we present two examples in Figure 5. We visualize the expert with the highest score as the routed expert for each instance pair. In the provided examples, we observe that expert-0 specialize in downward-related concepts, while expert-4 focuses on parabolic trend. These examples also demonstrate the interpretability of MoPE.

**Number of Experts.** In Figure 6, we set the learning rate to 0.0001 and conducted four sets of experiments on the ETTh1 and ETTh2 datasets, $K_t = 1$, $K_f = \{1, 2, 4, 8\}$, to explore the effect of the number of frequency experts on the results. For example, $K_t 1 K_f 4$ means that the TFPS contains 1 time experts and 4 frequency experts. We observed that $K_t 1 K_f 2$ outperformed $K_t 1 K_f 4$ in both cases, suggesting that increasing the number of experts does not always lead to better performance.

In addition, we conducted three experiments based on the optimal number of frequency experts to verify the impact of varying the number of time experts on the results. As shown in Figure 6, the best results for ETTh1 were obtained with $K_t 4 K_f 2$, while for ETTh2, the optimal results were achieved with $K_t 2 K_f 2$. Combined with the average MMD in Table 5 (Appendix A), we attribute this to the fact that, in cases where concept drift is more severe, such as ETTh1 in the time domain, more experts are needed, whereas fewer experts are sufficient when the drift is less severe.

## 5 CONCLUSION

In this paper, we propose a novel pattern-aware time series forecasting framework, TFPS, which incorporates a dual-domain mixture of pattern experts approach. Our TFPS framework aims to address the distribution shift across time series patches and effectively assigns pattern-specific experts to model them. Experimental results across eight diverse datasets demonstrate that TFPS surpasses state-of-the-art methods in both quantitative metrics and visualizations. Future work will focus on investigating evolving distribution shifts, particularly those introduced by the emergence of new patterns, such as unforeseen epidemics or outbreaks.

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

## A DATASET

We evaluate the performance of TFPS on eight widely used datasets, including four ETT datasets (ETTh1, ETTh2, ETTm1 and ETTm2), Exchange, Weather, Electricity, and ILI. This subsection provides a summary of the datasets:

- **ETT** [1] (Zhou et al., 2021) (Electricity Transformer Temperature) dataset contains two electric transformers, ETT1 and ETT2, collected from two separate counties. Each of them has two versions of sampling resolutions (15min & 1h). Thus, there are four ETT datasets: **ETTm1**, **ETTm2**, **ETTh1**, and **ETTh2**.

- **Exchange-Rate** [2] (Lai et al., 2018) the exchange-rate dataset contains the daily exchange rates of eight foreign countries including Australia, British, Canada, Switzerland, China, Japan, New Zealand, and Singapore ranging from 1990 to 2016.

- **Weather** [3] (Wu et al., 2021) dataset contains 21 meteorological indicators in Germany, such as humidity and air temperature.

- **Electricity** [4] (Wu et al., 2021) is a dataset that describes 321 customers' hourly electricity consumption.

- **ILI** [5] (Wu et al., 2021) dataset collects the number of patients and influenza-like illness ratio in a weekly frequency.

For the data split, we follow Zeng et al. (2023) and split the data into training, validation, and testing by a ratio of 6:2:2 for the ETT datasets and 7:1:2 for the others. Details are shown in Table 5. The best parameters are selected based on the lowest validation loss and then applied to the test set for performance evaluation.

Table 5: The statistics of the datasets.

| Datasets | Variates | Prediction Length | Timesteps | Granularity | Average MMD$^*$ (Time Domain) | Average MMD$^*$ (Frequency Domain) |
|---|---|---|---|---|---|---|
| ETTh1 | 7 | {96, 192, 336, 720} | 17,420 | 1 hour | 0.938 | 0.340 |
| ETTh2 | 7 | {96, 192, 336, 720} | 17,420 | 1 hour | 0.582 | 0.635 |
| ETTm1 | 7 | {96, 192, 336, 720} | 69,680 | 15 min | 1.371 | 0.328 |
| ETTm2 | 7 | {96, 192, 336, 720} | 69,680 | 15 min | 1.213 | 0.815 |
| Exchange-Rate | 8 | {96, 192, 336, 720} | 7,588 | 1 day | 0.805 | 0.485 |
| Weather | 21 | {96, 192, 336, 720} | 52,696 | 10 min | 0.129 | 0.236 |
| Electricity | 321 | {96, 192, 336, 720} | 26,304 | 1 hour | 0.026 | 0.005 |
| ILI | 7 | {24, 36, 48, 60} | 966 | 1 week | 0.125 | 0.234 |

$^*$ A large MMD indicates a more severe drift.

## B MAXIMUM MEAN DISCREPANCY

Maximum mean discrepancy (MMD) is a kernel-based statistical test used to determine whether given two distribution are the same. Given an $X$, the feature map $\phi$ transforms $X$ to an another space $\mathcal{H}$ such that $\phi(X) \in \mathcal{H}$. $\mathcal{H}$ is Reproducing Kernel Hilbert Space (RKHS) and we can leverage the kernel trick to compute inner products in $\mathcal{H}$:

$$X, Y \quad \text{such that} \quad k(X, Y) = \langle \phi(X), \phi(Y) \rangle_{\mathcal{H}}. \tag{13}$$

**Feature means.** The mean embeddings of a probability distribution $P$ is a feature map that transforms $\phi(X)$ into the mean of each coordinate of $\phi(X)$:

$$\mu_P(\phi(X)) = [\mathbb{E}[\phi(X_1)], \cdots, \mathbb{E}[\phi(X_m)]]^T. \tag{14}$$

---

[1] https://github.com/zhouhaoyi/ETDataset
[2] https://github.com/laiguokun/multivariate-time-series-data
[3] https://www.bgc-jena.mpg.de/wetter/
[4] https://archive.ics.uci.edu/ml/datasets/ElectricityLoadDiagrams20112014
[5] https://gis.cdc.gov/grasp/fluview/fluportaldashboard.html

The inner product of the mean embeddings of $X \sim P$ and $Y \sim Q$ can be written in terms of kernel function:

$$\langle \mu_P(\phi(X)), \mu_Q(\phi(Y)) \rangle_{\mathcal{H}} = \mathbb{E}_{P,Q}[\langle \phi(X), \phi(Y) \rangle_{\mathcal{H}}] = \mathbb{E}_{P,Q}[k(X,Y)]. \tag{15}$$

**Maximum mean discrepancy.** The MMD measures the distance between the mean embeddings of two samples, $X$ and $Y$, in the RKHS:

$$\text{MMD}^2(P,Q) = \|\mu_P - \mu_Q\|_{\mathcal{H}}^2, \tag{16}$$

For convenience we omit the $\phi(\cdot)$ terms. If we use the norm induced by the inner product such that $\|x\| = \sqrt{\langle x, x \rangle}$, the Eq. 16 becomes:

$$\text{MMD}^2(P,Q) = \langle \mu_p - \mu_Q, \mu_p - \mu_Q \rangle = \langle \mu_p, \mu_p \rangle - 2\langle \mu_p, \mu_Q \rangle + \langle \mu_Q, \mu_Q \rangle. \tag{17}$$

Using the Eq. 15, finally above expression becomes:

$$\text{MMD}^2(P,Q) = \mathbb{E}_P[k(X,X)] - 2\mathbb{E}_{P,Q}[k(X,Y)] + \mathbb{E}_Q[k(Y,Y)]. \tag{18}$$

**Empirical estimation of MMD.** In real-world applications, the underlying distribution are usually unknown. Thus, an empirical estimate of Eq. 18 can be used:

$$\text{MMD}^2(X,Y) = \frac{1}{m(m-1)} \sum_{i \neq j} k(x_i, x_j) - \frac{2}{mn} \sum_{i,j} k(x_i, x_j) + \frac{1}{n(n-1)} \sum_{i \neq j} k(y_i, y_j), \tag{19}$$

where $x_i$ and $x_j$ are samples from $P$, $y_i$ and $y_j$ are samples from $Q$, and $k(x,y)$ is the kernel function, often the Gaussian (RBF) kernel.

## C    DISTRIBUTION SHIFTS IN BOTH TIME AND FREQUENCY DOMAINS

The time series $\mathcal{X}$ is segmented into $N$ patches, where each patch $\mathcal{P}_n = \{x_{n1}, x_{n2}, \ldots, x_{nP}\}$ consists of $P$ consecutive timesteps for $n = 1, 2, \cdots, N$. For the frequency domain, we apply a Fourier transform $\mathcal{F}$ to each patch $\mathcal{P}_n$, obtaining its frequency-domain representation as $\hat{\mathcal{P}}_n = \mathcal{F}(\mathcal{P}_n)$.

Each patch's probability distribution in the time domain is denoted as $p_t(\mathcal{P}_n)$, representing the statistical properties of $\mathcal{P}_n$, while its frequency domain distribution, denoted as $p_f(\hat{\mathcal{P}}_n)$, captures its spectral characteristics.

The distribution shifts between two patches $\mathcal{P}_i$ and $\mathcal{P}_j$ are characterized by the comparing their probability distributions in both time and frequency domains. These shifts are defined as:

$$\mathcal{D}_t(\mathcal{P}_i, \mathcal{P}_j) = |d(p_t(\mathcal{P}_i), p_t(\mathcal{P}_j))| > \theta, \tag{20}$$

$$\mathcal{D}_f(\hat{\mathcal{P}}_i, \hat{\mathcal{P}}_j) = |d(p_f(\hat{\mathcal{P}}_i), p_f(\hat{\mathcal{P}}_j))| > \theta, \tag{21}$$

where $d$ is a distance metric, such as MMD values or Kullback-Leibler divergence, and $\theta$ is a threshold indicating a significant distribution shift. If $\mathcal{D}_t(\mathcal{P}_i, \mathcal{P}_j)$ or $\mathcal{D}_f(\hat{\mathcal{P}}_i, \hat{\mathcal{P}}_j)$ exceeds $\theta$, this implies a significant distribution shift between the two patches in either domain.

## D    RELATED WORK

**Mixture-of-Experts.** Mixture-of-Experts (MoE) models have gained attention for their ability to scale efficiently by activating only a subset of experts for each input, as first introduced by Shazeer et al. (2017). Despite their success, challenges such as training instability, expert redundancy, and limited expert specialization have been identified (Puigcerver et al., 2023; Dai et al., 2024). These issues hinder the full potential of MoE models in real-world tasks.

Recent advances have integrated MoE with Transformers to improve scalability and efficiency. For example, GLaM (Du et al., 2022) and Switch Transformer (Fedus et al., 2022) interleave MoE layers with Transformer blocks, reducing computational costs. Other models like state space models

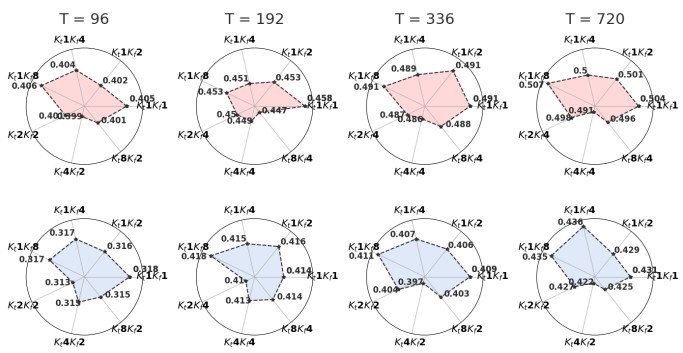

Figure 7: Results of expert number experiments for ETTh1 and ETTh2.

(SSMs) (Pióro et al., 2024; Anthony et al., 2024), (Alkilane et al., 2024) combines MoE with alternative architectures for enhanced scalability and inference speed.

In contrast, our approach introduces MoE into time series forecasting by assigning experts to specific time-frequency patterns, enabling more effective, patch-level adaptation. This approach represents a significant innovation in time series forecasting, offering a more targeted and effective way to handle varying patterns across both time and frequency domains.

# E    MORE MODEL ANALYSIS

## E.1    ANALYSIS OF EXPERTS

**Detailed Results on the Number of Experts.**

We provide the full results on the number of experts for the ETTh1 and ETTh2 dataset in Figure 7.

In Figure 6, we set the learning rate to 0.0001 and conducted four sets of experiments on the ETTh1 and ETTh2 datasets, $K_t = 1$, $K_f = \{1, 2, 4, 8\}$, to explore the effect of the number of frequency experts on the results. For example, $K_t 1 K_f 4$ means that the TFPS contains 1 time experts and 4 frequency experts. We observed that $K_t 1 K_f 2$ outperformed $K_t 1 K_f 4$ in both cases, suggesting that increasing the number of experts does not always lead to better performance.

In addition, we conducted three experiments based on the optimal number of frequency experts to verify the impact of varying the number of time experts on the results. As shown in Figure 7, the best results for ETTh1 were obtained with $K_t 4 K_f 2$, $K_t 8 K_f 4$, $K_t 4 K_f 4$, $K_t 4 K_f 4$, while for ETTh2, the optimal results were achieved with $K_t 2 K_f 2$, $K_t 2 K_f 4$, $K_t 4 K_f 2$ and $K_t 4 K_f 2$. Combined with the average MMD in Table 5, we attribute this to the fact that, in cases where concept drift is more severe, such as ETTh1 in the time domain, more experts are needed, whereas fewer experts are sufficient when the drift is less severe.

**Comparing Inter- and Intra-Cluster Differences via MMD.**

We present the heatmaps of inter-cluster and intra-cluster MMD values obtained using linear layers and PI in Figure 8. The diagonal elements represent the average MMD values of patches within the same clusters. If these values are small, it indicates that the difference of patches within the same cluster is relatively similar. The off-diagonal elements represent the average MMD values between patches from different clusters, where larger values mean significant differences between the clusters. We observe that when using PI, the intra-cluster drift is smaller, while the inter-cluster shift is more pronounced compared to the linear layer. This indicates that our identifier effectively classifies and distinguishes between different patterns.

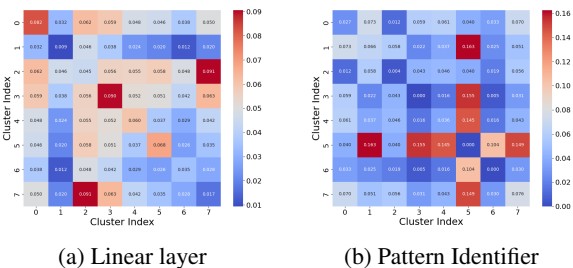

(a) Linear layer      (b) Pattern Identifier

Figure 8: Heatmap showing the MMD values of inter- and intra-cluster patches on ETTh1.

Table 6: Detailed results of the comparison between TFPS and normalization methods. The best results are highlighted in **bold** and the second best are underlined.

| Model | | IMP. | TFPS (Our) | | FEDformer | | | | | | | | | |
|---|---|---|---|---|---|---|---|---|---|---|---|---|---|---|
| | | | | | + SIN (2024a) | | + SAN (2023b) | | + Dish-TS (2023) | | + NST (2022) | | + RevIN (2021) | |
| Metric | | MSE | MSE | MAE | MSE | MAE | MSE | MAE | MSE | MAE | MSE | MAE | MSE | MAE |
| ETTh1 | 96 | -1.0% | 0.398 | 0.413 | 0.413 | **0.372** | **0.383** | 0.409 | 0.390 | 0.424 | 0.394 | 0.414 | 0.392 | 0.413 |
| | 192 | 3.8% | **0.423** | 0.423 | 0.443 | **0.417** | 0.431 | 0.438 | 0.441 | 0.458 | 0.441 | 0.442 | 0.443 | 0.444 |
| | 336 | -0.3% | 0.484 | 0.461 | **0.465** | **0.448** | 0.471 | 0.456 | 0.495 | 0.486 | 0.485 | 0.466 | 0.495 | 0.467 |
| | 720 | 4.5% | **0.488** | **0.476** | 0.509 | 0.490 | 0.504 | 0.488 | 0.519 | 0.509 | 0.505 | 0.496 | 0.520 | 0.498 |
| ETTh2 | 96 | 31.3% | 0.313 | **0.355** | 0.412 | 0.357 | **0.300** | 0.355 | 0.806 | 0.589 | 0.381 | 0.403 | 0.380 | 0.402 |
| | 192 | 26.0% | 0.405 | **0.410** | 0.472 | 0.453 | **0.392** | 0.413 | 0.936 | 0.659 | 0.478 | 0.453 | 0.457 | 0.443 |
| | 336 | 36.7% | **0.392** | **0.415** | 0.527 | 0.527 | 0.459 | 0.462 | 1.039 | 0.702 | 0.561 | 0.499 | 0.515 | 0.479 |
| | 720 | 37.9% | **0.410** | **0.433** | 0.593 | 0.639 | 0.462 | 0.472 | 1.237 | 0.759 | 0.502 | 0.481 | 0.507 | 0.487 |
| ETTm1 | 96 | 4.1% | 0.327 | 0.367 | 0.373 | **0.320** | **0.311** | 0.355 | 0.348 | 0.397 | 0.336 | 0.382 | 0.340 | 0.385 |
| | 192 | 2.9% | 0.374 | 0.395 | 0.394 | **0.366** | **0.351** | 0.383 | 0.406 | 0.428 | 0.386 | 0.409 | 0.390 | 0.411 |
| | 336 | 5.3% | 0.401 | 0.408 | 0.418 | **0.405** | **0.390** | 0.407 | 0.438 | 0.450 | 0.438 | 0.441 | 0.432 | 0.436 |
| | 720 | -0.5% | 0.479 | 0.456 | **0.451** | 0.475 | 0.456 | **0.444** | 0.497 | 0.481 | 0.483 | 0.460 | 0.497 | 0.466 |
| ETTm2 | 96 | 33.5% | **0.170** | 0.255 | 0.326 | **0.211** | 0.175 | 0.266 | 0.394 | 0.395 | 0.191 | 0.272 | 0.192 | 0.272 |
| | 192 | 32.3% | **0.235** | **0.296** | 0.402 | 0.316 | 0.246 | 0.315 | 0.552 | 0.472 | 0.270 | 0.321 | 0.270 | 0.320 |
| | 336 | 35.0% | **0.297** | **0.335** | 0.465 | 0.399 | 0.315 | 0.362 | 0.808 | 0.601 | 0.353 | 0.371 | 0.348 | 0.367 |
| | 720 | 35.9% | **0.401** | **0.397** | 0.555 | 0.547 | 0.412 | 0.422 | 1.282 | 0.771 | 0.445 | 0.422 | 0.430 | 0.415 |
| Weather | 96 | 28.4% | **0.154** | **0.202** | 0.280 | 0.215 | 0.179 | 0.239 | 0.244 | 0.317 | 0.187 | 0.234 | 0.187 | 0.234 |
| | 192 | 23.3% | **0.205** | **0.249** | 0.314 | 0.264 | 0.234 | 0.296 | 0.320 | 0.380 | 0.235 | 0.272 | 0.235 | 0.272 |
| | 336 | 19.8% | **0.262** | **0.289** | 0.329 | 0.293 | 0.304 | 0.348 | 0.424 | 0.452 | 0.289 | 0.308 | 0.287 | 0.307 |
| | 720 | 18.4% | **0.344** | **0.342** | 0.382 | 0.370 | 0.400 | 0.404 | 0.604 | 0.553 | 0.359 | 0.352 | 0.361 | 0.353 |
| 1st (2nd) Count | | | 24 (8) | | 9 (4) | | 7 (24) | | 0 (1) | | 0 (1) | | 0 (2) | |

## E.2 RESULTS OF THE COMPARISON BETWEEN TFPS AND NORMALIZATION METHODS

In this section, we provide the detailed experimental results of the comparison between TFPS and five state-of-the-art normalization methods for non-stationary time series forecasting: SIN (Han et al., 2024a), SAN (Liu et al., 2023b), Dish-TS (Fan et al., 2023), Non-Stationary Transformers (NST) (Liu et al., 2022), and RevIN (Kim et al., 2021). The results of SIN are from Han et al. (2024a), other results are from Liu et al. (2023b). We report the evaluation of FEDformer over all the forecasting lengths for each dataset and the relative improvements in Table 6. It can be concluded that TFPS achieves the best performance among existing methods in most cases. The improvement is significant with an average MSE decrease of 18.9%. We attribute this improvement to the accurate identification of pattern groups and the provision of specialized experts for each group, thereby avoiding the over-stationarization problem often associated with normalization methods.

## F   METRIC ILLUSTRATION

We use mean square error (MSE) and mean absolute error (MAE) as our metrics for evaluation of all forecasting models. Then calculation of MSE and MAE can be described as:

$$\text{MSE} = \frac{1}{H} \sum_{i=L+1}^{L+H} (\hat{Y}_i - Y_i)^2, \tag{22}$$

$$\text{MAE} = \frac{1}{H} \sum_{i=L+1}^{L+H} \left| \hat{Y}_i - Y_i \right|, \tag{23}$$

where $\hat{Y}$ is predicted vector with $H$ future values, while $Y$ is the ground truth.

## G   ALGORITHM OF TFPS

We provide the pseudo-code of TFPS in Algorithm 1.

## H   BROADER IMPACT

**Real-world applications.** TFPS addresses the crucial challenge of time series forecasting, which is a valuable and urgent demand in extensive applications. Our method achieves consistent state-of-the-art performance in four real-world applications: electricity, weather, exchange rate, illness. Researchers in these fields stand to benefit significantly from the enhanced forecasting capabilities of TFPS. We believe that improved time series forecasting holds the potential to empower decision-making and proactively manage risks in a wide array of societal domains.

**Academic research.** TFPS draws inspiration from classical time series analysis and stochastic process theory, contributing to the field by introducing a novel framework with the assistance pattern recognition. This innovative architecture and its associated methodologies represent significant advancements in the field of time series forecasting, enhancing the model's ability to address distribution shifts and complex patterns effectively.

**Model Robustness.** Extensive experimentation with TFPS reveals robust performance without exceptional failure cases. Notably, TFPS exhibits impressive results and maintains robustness in datasets with distribution shifts. The pattern identifier structure within TFPS groups the time series into distinct patterns and adopts a mixture of pattern experts for further prediction, thereby alleviating prediction difficulties. However, it is essential to note that, like any model, TFPS may face challenges when dealing with unpredictable patterns, where predictability is inherently limited. Understanding these nuances is crucial for appropriately applying and interpreting TFPS's outcomes.

Our work only focuses on the scientific problem, so there is no potential ethical risk.

## I   LIMITATIONS

Though TFPS demonstrates promising performance on the benchmark dataset, there are still some limitations of this method. First, the patch length is primarily chosen heuristically, and the current design struggles with handling indivisible lengths or multi-period characteristics in time series. While this approach works well in experiments, it lacks generalizability for real-world applications. Second, the real-world time series data undergo expansion, implying that the new patterns continually emerge over time, such as an epidemic or outbreak that had not occurred before. Therefore, future work will focus on developing a more flexible and automatic patch length selection mechanism, as well as an extensible solution to address these evolving distribution shifts.

---

**Algorithm 1** Time-Frequency Pattern-Specific architecture - Overall Architecture.

---

**Input**: Input lookback time series $X \in \mathbb{R}^{L \times C}$; input length $L$; predicted length $H$; variables number $C$; patch length $P$; feature dimension $D$; encoder layers number $n$; random Gaussian distribution-initialized subspace $\mathbf{D} = [\mathbf{D}^{(1)}, \mathbf{D}^{(2)}, \cdots, \mathbf{D}^{(K)}]$, each $\mathbf{D}^{(j)} \in \mathbf{R}^{q \times d}$, where $q = C \times D$ and $d = q/K$. Technically, we set $D$ as 512, $n$ as 2.

**Output**: The prediction result $\hat{Y}$.

1:  $X = X.\texttt{transpose}$       $\triangleright X \in \mathbb{R}^{C \times L}$

2:  $X_{PE} = \texttt{Patch}(X) + \texttt{Position Embedding}$    $\triangleright X_t^0 \in \mathbb{R}^{C \times N \times D}$

3:  $\triangleright$ Time Encoder.

4:  $X_t^0 = X_{PE}$

5:  **for** $l$ **in** $\{1, \ldots, n\}$**:**

6:      $X_t^{l-1} = \texttt{LayerNorm}(X_t^{l-1} + \texttt{Self-Attn}(X_t^{l-1}))$.    $\triangleright X_t^{l-1} \in \mathbb{R}^{C \times N \times D}$

7:      $X_t^l = \texttt{LayerNorm}(X_t^{l-1} + \texttt{Feed-Forward}(X_t^{l-1}))$.    $\triangleright X_t^l \in \mathbb{R}^{C \times N \times D}$

8:  **End for**

9:  $z_t = X_t^l$       $\triangleright z_t^l \in \mathbb{R}^{C \times N \times D}$

10:  $\triangleright$ Pattern Identifier for Time Domain.

11:  $s_t = \texttt{Subspace affinity}(z_t, \mathbf{D})$    $\triangleright$ Eq. 6 of the paper $s_t \in \mathbb{R}^{C \times N \times D}$

12:  $\widetilde{s}_t = \texttt{Subspace refinement}(s_t)$    $\triangleright$ Eq. 7 of the paper $\widetilde{s}_t \in \mathbb{R}^{C \times N \times D}$

13:  $\triangleright$ Mixture of Temporal Pattern Experts.

14:  $G(s) = \texttt{Softmax}(\texttt{TopK}(s_t))$

15:  $h_t = \sum_{k=1}^{K} G(s)\texttt{MLP}_k(z_t)$    $\triangleright$ Eq. 10 and Eq. 11 of the paper $h_t \in \mathbb{R}^{C \times N \times D}$

16:  $\triangleright$ Frequency Encoder.

17:  $X_f^0 = X_{PE}$       $\triangleright$ Eq. 2 of the paper $X_f^0 \in \mathbb{R}^{C \times N \times P}$

18:  **for** $l$ **in** $\{1, \ldots, n\}$**:**

19:      $X_f^{l-1} = \texttt{LayerNorm}(X_f^{l-1} + \texttt{Fourier}(X_f^{l-1}))$.    $\triangleright X_f^{l-1} \in \mathbb{R}^{C \times N \times D}$

20:      $X_f^l = \texttt{LayerNorm}(X_f^{l-1} + \texttt{Feed-Forward}(X_f^{l-1}))$.    $\triangleright X_f^l \in \mathbb{R}^{C \times N \times D}$

21:  **End for**

22:  $z_f = X_f^l$       $\triangleright z_f^n \in \mathbb{R}^{C \times N \times D}$

23:  $\triangleright$ Pattern Identifier for Frequency Domain.

24:  $s_f = \texttt{Subspace affinity}(z_f, \mathbf{D})$    $\triangleright$ Eq. 6 of the paper $s_f \in \mathbb{R}^{C \times N \times D}$

25:  $\widetilde{s}_f = \texttt{Subspace refinement}(s_f)$    $\triangleright$ Eq. 7 of the paper $\widetilde{s}_f \in \mathbb{R}^{C \times N \times D}$

26:  $\triangleright$ Mixture of Frequency Pattern Experts.

27:  $G(s) = \texttt{Softmax}(\texttt{TopK}(s_f))$

28:  $h_f = \sum_{k=1}^{K} G(s)\texttt{MLP}_k(z_f)$    $\triangleright$ Eq. 10 and Eq. 11 of the paper $h_f \in \mathbb{R}^{C \times N \times D}$

29:  $h = \texttt{Concat}(h_t, h_f)$       $\triangleright h \in \mathbb{R}^{C \times N \times 2*D}$

30:  **for** $c$ **in** $\{1, \ldots, C\}$**:**

31:      $\hat{Y} = \texttt{Linear}(\texttt{Flatten}(h))$.    $\triangleright$ Project tokens back to predicted series $\hat{Y} \in \mathbb{R}^{C \times H}$

32:  **End for**

33:  $\hat{Y} = \hat{Y}.\texttt{transpose}$       $\triangleright \hat{Y} \in \mathbb{R}^{H \times C}$

34:  **Return** $\hat{Y}$    $\triangleright$ Output the final prediction $\hat{Y} \in \mathbb{R}^{H \times C}$

---

Table 7: Multivariate long-term forecasting results for Traffic. The input lengths is $L = 96$. The best results are highlighted in **bold** and the second best are underlined.

| Model | IMP. | TFPS (Our) | | TSLANet (2024) | | FITS (2024) | | iTransformer (2024a) | | TFDNet-IK (2023) | | PatchTST (2023) | | TimesNet (2023a) | | DLinear (2023) | | FEDformer (2022) | |
|---|---|---|---|---|---|---|---|---|---|---|---|---|---|---|---|---|---|---|---|
| Metric | MSE | MSE | MAE | MSE | MAE | MSE | MAE | MSE | MAE | MSE | MAE | MSE | MAE | MSE | MAE | MSE | MAE | MSE | MAE |
| Traffic 96 | 21.1% | **0.427** | 0.296 | 0.475 | 0.307 | 0.651 | 0.388 | 0.428 | **0.295** | 0.519 | 0.314 | 0.446 | 0.284 | 0.586 | 0.316 | 0.650 | 0.397 | 0.575 | 0.357 |
| 192 | 17.7% | **0.445** | **0.298** | 0.478 | 0.306 | 0.603 | 0.364 | 0.448 | 0.302 | 0.513 | 0.314 | 0.453 | 0.285 | 0.618 | 0.323 | 0.600 | 0.372 | 0.613 | 0.381 |
| 336 | 17.0% | **0.459** | **0.307** | 0.494 | 0.312 | 0.610 | 0.366 | 0.465 | 0.311 | 0.525 | 0.319 | 0.467 | 0.291 | 0.634 | 0.337 | 0.606 | 0.374 | 0.622 | 0.380 |
| 720 | 15.1% | **0.496** | **0.313** | 0.528 | 0.331 | 0.648 | 0.387 | 0.501 | 0.333 | 0.561 | 0.336 | 0.501 | 0.492 | 0.659 | 0.349 | 0.646 | 0.396 | 0.630 | 0.383 |
| 1st Count | | 7 | | 0 | | 0 | | 1 | | 0 | | 0 | | 0 | | 0 | | 0 | |

Table 8: Experiment results under hyperparameter searching for the long-term forecasting task. The best results are highlighted in **bold** and the second best are underlined.

| Model | IMP. | TFPS (Our) | | TSLANet (2024) | | FITS (2024) | | iTransformer (2024a) | | TFDNet-IK (2023) | | PatchTST (2023) | | TimesNet (2023a) | | Dlinear (2023) | | FEDformer (2022) | |
|---|---|---|---|---|---|---|---|---|---|---|---|---|---|---|---|---|---|---|---|
| Metric | MSE | MSE | MAE | MSE | MAE | MSE | MAE | MSE | MAE | MSE | MAE | MSE | MAE | MSE | MAE | MSE | MAE | MSE | MAE |
| ETTh1 96 | 1.5% | 0.372 | 0.404 | 0.368 | 0.394 | 0.374 | 0.395 | 0.387 | 0.405 | **0.360** | **0.387** | 0.375 | 0.400 | 0.389 | 0.412 | 0.384 | 0.405 | 0.385 | 0.425 |
| 192 | 5.7% | **0.401** | **0.410** | 0.413 | 0.418 | 0.407 | 0.414 | 0.441 | 0.436 | 0.403 | 0.412 | 0.414 | 0.421 | 0.441 | 0.442 | 0.443 | 0.450 | 0.441 | 0.461 |
| 336 | 9.8% | **0.409** | **0.402** | 0.412 | 0.416 | 0.429 | 0.428 | 0.491 | 0.463 | 0.434 | 0.429 | 0.432 | 0.436 | 0.491 | 0.467 | 0.447 | 0.448 | 0.491 | 0.473 |
| 720 | 11.2% | **0.423** | **0.433** | 0.473 | 0.477 | 0.425 | 0.446 | 0.509 | 0.494 | 0.437 | 0.452 | 0.450 | 0.466 | 0.512 | 0.491 | 0.504 | 0.515 | 0.501 | 0.499 |
| ETTh2 96 | 9.3% | **0.268** | **0.325** | 0.283 | 0.344 | 0.274 | 0.337 | 0.301 | 0.350 | 0.271 | 0.329 | 0.278 | 0.336 | 0.324 | 0.368 | 0.290 | 0.353 | 0.342 | 0.383 |
| 192 | 10.4% | **0.329** | 0.376 | 0.331 | 0.378 | 0.337 | 0.377 | 0.380 | 0.399 | 0.333 | 0.372 | 0.339 | 0.380 | 0.393 | 0.410 | 0.388 | 0.422 | 0.434 | 0.440 |
| 336 | 17.7% | 0.329 | 0.401 | 0.319 | **0.377** | 0.360 | 0.398 | 0.424 | 0.432 | 0.361 | 0.396 | 0.336 | 0.380 | 0.429 | 0.437 | 0.463 | 0.473 | 0.512 | 0.497 |
| 720 | 9.0% | 0.412 | 0.441 | 0.407 | 0.449 | 0.386 | 0.423 | 0.430 | 0.447 | 0.382 | 0.418 | 0.382 | 0.421 | 0.433 | 0.448 | 0.733 | 0.606 | 0.467 | 0.476 |
| ETTm1 96 | 10.2% | **0.281** | 0.329 | 0.291 | 0.353 | 0.303 | 0.345 | 0.342 | 0.377 | 0.283 | 0.330 | 0.288 | 0.342 | 0.337 | 0.377 | 0.301 | 0.345 | 0.360 | 0.406 |
| 192 | 8.5% | **0.324** | **0.354** | 0.329 | 0.372 | 0.337 | 0.365 | 0.383 | 0.396 | 0.327 | 0.356 | 0.334 | 0.372 | 0.395 | 0.406 | 0.336 | 0.366 | 0.395 | 0.427 |
| 336 | 8.2% | 0.359 | 0.404 | 0.372 | 0.392 | 0.372 | 0.385 | 0.418 | 0.418 | 0.361 | 0.375 | 0.367 | 0.393 | 0.433 | 0.432 | 0.372 | 0.389 | 0.448 | 0.458 |
| 720 | 8.2% | **0.409** | **0.408** | 0.423 | 0.425 | 0.428 | 0.416 | 0.487 | 0.457 | 0.411 | 0.409 | 0.417 | 0.422 | 0.484 | 0.458 | 0.427 | 0.423 | 0.491 | 0.479 |
| ETTm2 96 | 8.9% | **0.158** | **0.243** | 0.167 | 0.256 | 0.165 | 0.255 | 0.186 | 0.272 | 0.158 | 0.244 | 0.164 | 0.253 | 0.182 | 0.262 | 0.172 | 0.267 | 0.193 | 0.285 |
| 192 | 5.7% | 0.222 | 0.302 | 0.221 | 0.294 | 0.220 | 0.291 | 0.254 | 0.314 | 0.219 | 0.282 | 0.221 | 0.292 | 0.252 | 0.307 | 0.237 | 0.314 | 0.256 | 0.324 |
| 336 | 8.5% | **0.268** | **0.316** | 0.277 | 0.329 | 0.274 | 0.326 | 0.316 | 0.351 | 0.273 | 0.317 | 0.277 | 0.329 | 0.312 | 0.346 | 0.295 | 0.359 | 0.321 | 0.364 |
| 720 | 12.0% | **0.344** | **0.373** | 0.356 | 0.382 | 0.367 | 0.383 | 0.414 | 0.407 | 0.346 | 0.374 | 0.365 | 0.384 | 0.417 | 0.404 | 0.427 | 0.439 | 0.434 | 0.426 |
| Traffic 96 | 17.8% | **0.370** | 0.257 | 0.375 | 0.260 | 0.398 | 0.285 | 0.428 | 0.295 | 0.377 | 0.253 | OOM | | 0.586 | 0.316 | 0.413 | 0.287 | 0.575 | 0.357 |
| 192 | 17.0% | **0.391** | 0.269 | 0.395 | 0.272 | 0.408 | 0.288 | 0.448 | 0.302 | 0.391 | 0.260 | | | 0.618 | 0.323 | 0.424 | 0.290 | 0.613 | 0.381 |
| 336 | 17.2% | **0.401** | 0.271 | 0.402 | 0.272 | 0.420 | 0.292 | 0.465 | 0.311 | 0.408 | 0.266 | | | 0.634 | 0.337 | 0.438 | 0.299 | 0.622 | 0.380 |
| 720 | 15.7% | 0.432 | 0.294 | 0.431 | 0.288 | 0.448 | 0.310 | 0.501 | 0.333 | 0.451 | 0.291 | | | 0.659 | 0.349 | 0.466 | 0.316 | 0.630 | 0.383 |
| Electricity 96 | 10.3% | 0.134 | 0.225 | 0.137 | 0.229 | 0.135 | 0.231 | 0.148 | 0.239 | **0.130** | 0.222 | 0.130 | 0.223 | 0.168 | 0.272 | 0.140 | 0.237 | 0.188 | 0.303 |
| 192 | 11.9% | **0.145** | 0.238 | 0.153 | 0.242 | 0.149 | 0.244 | 0.167 | 0.258 | 0.146 | 0.237 | 0.149 | 0.240 | 0.186 | 0.289 | 0.154 | 0.250 | 0.197 | 0.311 |
| 336 | 6.8% | 0.166 | 0.258 | 0.165 | 0.242 | 0.165 | 0.260 | 0.178 | 0.271 | 0.162 | 0.254 | 0.168 | 0.262 | 0.196 | 0.297 | 0.169 | 0.268 | 0.212 | 0.327 |
| 720 | 6.9% | **0.200** | 0.291 | 0.206 | 0.294 | 0.204 | 0.293 | 0.211 | 0.300 | 0.201 | 0.287 | 0.204 | 0.289 | 0.235 | 0.329 | 0.204 | 0.300 | 0.243 | 0.352 |
| 1st Count | | 26 | | 5 | | 0 | | 0 | | 16 | | 1 | | 0 | | 0 | | 0 | |

## J  TRAFFIC RESULTS

We conducted addition experiments on high-dimensional Traffic dataset to further evaluate the performance and generalizability of TFPS, as shown in Table 7.

## K  HYPERPARAMETER-SEARCH RESULTS

To ensure a fair comparison between models, we conducted experiments using unified parameters $L = 96$ and reported results in the main text.

In addition, considering that the reported results in different papers are mostly obtained through hyperparameter search, we provide the experiment results with the full version of the parameter search. We searched for input length among 96, 192, 336, and 512. The results are included in Table 8. All baselines are reproduced by their official code.

We can find that the relative promotion of TFPS over TFDNet is smaller under comprehensive hyperparameter search than the unified hyperparameter setting. It is worth noticing that TFPS runs much faster than TFDNet according to the efficiency comparison in Table 11. Therefore, considering performance, hyperparameter-search cost and efficiency, we believe TFPS is a practical model in real-world applications and is valuable to deep time series forecasting community.

## L  VISUALIZATION OF CLUSTERING

Figure 9 presents the t-SNE visualization of the learned embedded representation on the ETTh1. In the Figure 9 (a), where the pattern identifier is replaced with a linear layer, the representation lacks

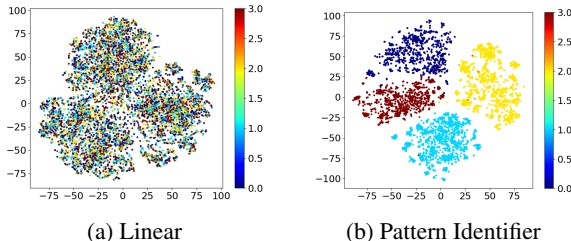

(a) Linear        (b) Pattern Identifier

Figure 9: Visualization of the embedded representations with t-SNE on ETTh1. The left figure shows the visualization when the Patch Identifier is replaced with a Linear Layer for comparison, while the right figure shows the visualization of the proposed method.

Table 9: Comparison between TFPS and MoE-based methods. The best results are highlighted in **bold** and the second best are underlined.

| Model | | IMP. | TFPS (Our) | | MoLE 2024 | | MoU 2024 | | KAN4TSF 2024b | |
|---|---|---|---|---|---|---|---|---|---|---|
| Metric | | MSE | MSE | MAE | MSE | MAE | MSE | MAE | MSE | MAE |
| ETTh1 | 96 | -4.3% | 0.398 | 0.413 | 0.383 | **0.392** | **0.381** | 0.403 | 0.382 | 0.400 |
| | 192 | 1.7% | **0.423** | **0.423** | 0.434 | 0.426 | 0.429 | 0.430 | 0.430 | 0.426 |
| | 336 | 1.6% | **0.484** | **0.461** | 0.489 | 0.478 | 0.488 | 0.463 | 0.498 | 0.467 |
| | 720 | 8.2% | **0.488** | **0.476** | 0.602 | 0.545 | 0.499 | 0.484 | 0.494 | 0.479 |
| ETTh2 | 96 | 10.4% | **0.313** | **0.355** | 0.413 | 0.360 | 0.317 | 0.358 | 0.318 | 0.358 |
| | 192 | 10.3% | **0.405** | **0.410** | 0.525 | 0.416 | 0.409 | 0.414 | 0.419 | 0.414 |
| | 336 | 7.1% | **0.392** | **0.415** | 0.423 | 0.434 | 0.397 | 0.420 | 0.447 | 0.452 |
| | 720 | 8.4% | **0.410** | **0.433** | 0.453 | 0.458 | 0.412 | 0.434 | 0.477 | 0.476 |
| ETTm1 | 96 | 13.5% | **0.327** | **0.367** | 0.338 | 0.380 | 0.465 | 0.442 | 0.333 | 0.371 |
| | 192 | 10.6% | **0.374** | **0.395** | 0.388 | 0.403 | 0.483 | 0.455 | 0.384 | 0.399 |
| | 336 | 11.8% | **0.401** | **0.408** | 0.417 | 0.431 | 0.540 | 0.488 | 0.407 | 0.413 |
| | 720 | 7.3% | **0.479** | **0.456** | 0.486 | 0.472 | 0.583 | 0.509 | 0.483 | 0.469 |
| ETTm2 | 96 | 13.9% | **0.170** | **0.255** | 0.238 | 0.271 | 0.179 | 0.263 | 0.175 | 0.260 |
| | 192 | 3.8% | **0.235** | **0.296** | 0.247 | 0.305 | 0.243 | 0.303 | 0.244 | 0.305 |
| | 336 | 3.3% | **0.297** | **0.335** | 0.308 | 0.343 | 0.306 | 0.343 | 0.308 | 0.347 |
| | 720 | 13.7% | **0.401** | **0.397** | 0.583 | 0.419 | 0.405 | 0.404 | 0.405 | 0.404 |
| 1ˢᵗ Count | | | 30 | | 1 | | 1 | | 0 | |

clear clustering structures, resulting in scattered and indistinct groupings. In contrast, Figure 9 (b) shows the visualization of the representation learned by the proposed method, which effectively captures discriminative features and reveals significantly clearer clustering patterns.

# M    COMPARED WITH MOE-BASED METHODS

As shown in Table 9, unlike MoE-based methods that rely on the Softmax function as a gating mechanism, our approach constructs a pattern recognizer to assign different experts to handle distinct patterns. This results in TFPS achieving relative improvements of 2.3%, 9.0%, 10.6%, and 9.1% across the four datasets, respectively.

# N    COMPARED WITH DISTRIBUTION SHIFT METHODS

As shown in Table 10, we compare with the methods for distribution shift. This results in TFPS achieving relative improvements of 6.7%, 6.6%, 4.8%, and 5.9% across the four datasets, respectively.

# O    EFFICIENCY ANALYSIS

To make this clearer, we present the results of ETTh1 for a prediction length of 192 from Table 2 and include additional results on runtime and computational complexity in Table 11. Due to the sparsity of MoPE, TFPS achieves a balance between performance and efficiency:

Table 10: Comparison between TFPS and methods for Distribution Shift. The best results are highlighted in **bold** and the second best are underlined.

| Model | | IMP. | TFPS (Our) | | Koopa 2024b | | SOLID 2024a | | OneNet 2024 | |
|---|---|---|---|---|---|---|---|---|---|---|
| Metric | | MSE | MSE | MAE | MSE | MAE | MSE | MAE | MSE | MAE |
| ETTh1 | 96 | 7.9% | 0.398 | 0.413 | **0.385** | 0.407 | 0.440 | 0.439 | 0.425 | **0.402** |
| | 192 | 10.3% | **0.423** | **0.423** | 0.445 | 0.434 | 0.492 | 0.466 | 0.452 | 0.443 |
| | 336 | 4.9% | **0.484** | **0.461** | 0.489 | 0.460 | 0.525 | 0.481 | 0.492 | 0.482 |
| | 720 | 4.4% | **0.488** | **0.476** | 0.497 | 0.480 | 0.517 | 0.496 | 0.504 | 0.496 |
| ETTh2 | 96 | 10.6% | **0.313** | **0.355** | 0.318 | 0.360 | 0.318 | 0.359 | 0.382 | 0.362 |
| | 192 | 4.7% | 0.405 | 0.410 | **0.378** | **0.398** | 0.414 | 0.418 | 0.435 | 0.426 |
| | 336 | 4.8% | **0.392** | **0.415** | 0.415 | 0.430 | 0.398 | 0.421 | 0.426 | 0.419 |
| | 720 | 6.8% | **0.410** | **0.433** | 0.445 | 0.456 | 0.424 | 0.441 | 0.456 | 0.437 |
| ETTm1 | 96 | 6.8% | **0.327** | **0.367** | 0.329 | 0.359 | 0.329 | 0.370 | 0.374 | 0.392 |
| | 192 | 2.0% | **0.374** | 0.395 | 0.380 | **0.393** | 0.379 | 0.400 | 0.385 | 0.435 |
| | 336 | 8.7% | **0.401** | **0.408** | 0.401 | 0.411 | 0.405 | 0.412 | 0.473 | 0.458 |
| | 720 | 2.0% | 0.479 | 0.456 | **0.475** | **0.448** | 0.482 | 0.464 | 0.496 | 0.483 |
| ETTm2 | 96 | 5.3% | **0.170** | **0.255** | 0.179 | 0.261 | 0.175 | 0.258 | 0.184 | 0.274 |
| | 192 | 3.8% | **0.235** | **0.296** | 0.246 | 0.305 | 0.241 | 0.302 | 0.248 | 0.384 |
| | 336 | 3.4% | **0.297** | **0.335** | 0.310 | 0.348 | 0.303 | 0.342 | 0.313 | 0.374 |
| | 720 | 9.0% | **0.401** | **0.397** | 0.405 | 0.402 | 0.456 | 0.436 | 0.425 | 0.438 |
| 1st Count | | | 25 | | 6 | | 0 | | 1 | |

Table 11: The GPU memory (MB) and speed (inference time) of each model.

| | TFPS | TSLANet | FITS | iTransformer | TFDNet-IK | PatchTST | TimesNet | DLinear | FEDformer |
|---|---|---|---|---|---|---|---|---|---|
| MSE | **0.423** | 0.448 | 0.445 | 0.441 | 0.458 | 0.460 | 0.441 | 0.434 | 0.441 |
| GPU Memory (MB) | 9.643 | 0.481 | 0.019 | 3.304 | 0.246 | 0.205 | 2.345 | 0.142 | 62.191 |
| Average Inference Time (ms) | 6.457 | 2.100 | 1.202 | 2.949 | 407.853 | 17.851 | 72.196 | 0.789 | 259.001 |

**Performance Superiority**: TFPS achieves an MSE of 0.423, outperforming TSLANet (0.448), FITS (0.445), PatchTST (0.460), and FEDformer (0.441). This represents a 5.6% improvement over TSLANet and a 8.0% improvement over PatchTST, highlighting its significant accuracy gains. While DLinear achieves an MSE of 0.434, TFPS still demonstrates a 2.5% relative improvement, making it the most accurate model among all baselines.

**Efficiency Gains**: TFPS maintains competitive runtime and memory efficiency.

- Runtime: TFPS runs in 6.457 ms, making it 2.8× faster than PatchTST (17.851 ms) and 11.2× faster than TimesNet (72.196 ms).

- Memory Usage: TFPS uses 9.643 MB of GPU memory, significantly less than FEDformer (62.191 MB) and comparable to iTransformer (3.304 MB). This makes TFPS suitable for resource-constrained applications while maintaining superior performance.

**Balancing Trade-offs**: While lightweight models like DLinear (0.434 MSE, 0.789 ms runtime) are slightly more efficient, TFPS delivers a performance improvement of 2.5%, providing a well-rounded solution that balances accuracy and efficiency effectively.

# P HYPERPARAMETER SENSITIVITY

In this section, we analysis the impact of the hyperparameters $\alpha$ and $\beta$ on the performance.

Specifically, we performed a grid search to optimize the hyperparameters $\alpha_t = \{0.0001, 0.001, 0.01\}$ and $\alpha_f = \{0.0001, 0.001, 0.01\}$, as shown in Figure 10 (a). After extensive testing, we ultimately fixed at $\alpha_t = \alpha_f = 10^{-3}$ in our experiments.

In addition, we conducted a grid search to optimize the balance factors $\beta_t = \{0.01, 0.05, 0.1, 0.5, 1\}$ and $\beta_f = \{0.01, 0.05, 0.1, 0.5, 1\}$. The performance under different parameter values is displayed in Figure 10 (b), from which we have the following observations:

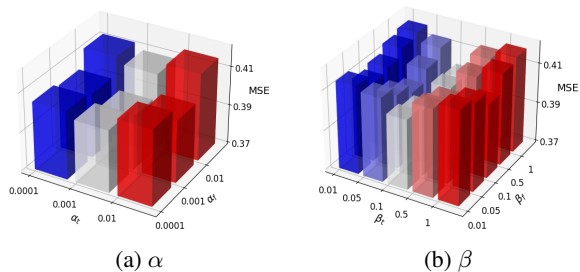

(a) $\alpha$                                      (b) $\beta$

Figure 10: Parameter sensitivity of $\alpha$ and $\beta$ of the proposed method on the ETTh1-96 dataset.

Table 12: In the table, w/ Imaginary indicates that we incorporate both the real and imaginary parts into the network.

|  | ETTh1 | | | | ETTh2 | | | |
| --- | --- | --- | --- | --- | --- | --- | --- | --- |
|  | 96 | 192 | 336 | 720 | 96 | 192 | 336 | 720 |
| TFPS | 0.398 | **0.423** | **0.484** | 0.488 | 0.313 | **0.405** | 0.392 | 0.410 |
| w/ Imaginary | **0.397** | 0.424 | 0.487 | **0.486** | **0.312** | 0.406 | **0.391** | **0.399** |

- Firstly, the performance is affected when the value of $\beta$ is too low, indicating that the proposed clustering objective plays a crucial role in distinguishing patterns.

- Second, an excessive $\beta$ also has a negative on the performance. One plausible explanation is that the excessive value influences the learning of the inherent structure of original data, resulting in a perturbation of the embedding space.

- Overall, we recommend setting $\beta$ around 0.1 for optimal performance.

# Q    FULL ABLATION

## Q.1    IMPACTS OF REAL/IMAGINARY PARTS

To further validate the robustness of our approach, we adopted similar operations in FreTS to conduct experiments incorporating both the real and imaginary parts. The results in the Table 12 show that the performance of TFPS with the real part only is very similar to that when both parts are included, while requiring fewer parameters. This further reinforces the conclusion that TFPS remains highly effective even when focusing solely on the real part of the Fourier transform.

## Q.2    ABLATION ON PI

The PI module plays a crucial role in identifying and characterizing distinct patterns within the time series data, while the gating network dynamically selects the most relevant experts for each segment. This collaborative mechanism allows the model to specialize in handling different patterns and adapt effectively to distribution shifts, thus mitigating the overfitting risks that arise from treating all data equally.

To validate the importance of PI empirically, we have conducted the ablation experiments comparing the model's performance by replacing the PI module with a linear layer in the Table 3 of main text. In addition, we supplement some ablation experiments in Table 13 to further verify the effectiveness of PI.

## Q.3    ABLATION ON $R_1$ AND $R_2$

We conducted ablation experiments to further verify the important roles of $R_1$ and $R_2$, as shown in Table 14.

Table 13: Ablation study of PI components. The model variants in our ablation study include the following configurations across both time and frequency branches: (a) inclusion of the Time PI; (b) inclusion of the Frequency PI; (c) exclusion of both. The best results are in **bold**.

| Time PI | Frequency PI | ETTh1 | | | | ETTh2 | | | |
|---|---|---|---|---|---|---|---|---|---|
| | | 96 | 192 | 336 | 720 | 96 | 192 | 336 | 720 |
| ✓ | ✓ | **0.398** | **0.423** | **0.484** | **0.488** | **0.313** | **0.405** | **0.392** | **0.410** |
| ✓ | ✗ | 0.404 | 0.454 | 0.490 | 0.503 | 0.322 | 0.413 | 0.410 | 0.425 |
| ✗ | ✓ | 0.405 | 0.456 | 0.493 | 0.509 | 0.324 | 0.415 | 0.412 | 0.430 |
| ✗ | ✗ | 0.407 | 0.458 | 0.497 | 0.513 | 0.328 | 0.418 | 0.419 | 0.435 |

Table 14: Ablation study of Loss Constraint. The model variants in our ablation study include the following configurations across both time and frequency branches: (a) inclusion of the $R_1$; (b) inclusion of the $R_2$; (c) exclusion of both. The best results are in **bold**.

| $R_1$ | $R_2$ | ETTh1 | | | | ETTh2 | | | |
|---|---|---|---|---|---|---|---|---|---|
| | | 96 | 192 | 336 | 720 | 96 | 192 | 336 | 720 |
| ✓ | ✓ | **0.398** | **0.423** | **0.484** | **0.488** | **0.313** | **0.405** | **0.392** | **0.410** |
| ✓ | ✗ | 0.408 | 0.449 | 0.500 | 0.498 | 0.320 | 0.418 | 0.415 | 0.429 |
| ✗ | ✓ | 0.403 | 0.434 | 0.493 | 0.491 | 0.316 | 0.413 | 0.405 | 0.418 |
| ✗ | ✗ | 0.412 | 0.456 | 0.509 | 0.503 | 0.328 | 0.425 | 0.420 | 0.435 |

Table 15: Multi-output predictor and a stacked attention layer are used to replace MoPE in ETTh1 and ETTh2 datasets.

| | ETTh1 | | | | ETTh2 | | | |
|---|---|---|---|---|---|---|---|---|
| | 96 | 192 | 336 | 720 | 96 | 192 | 336 | 720 |
| TFPS | **0.398** | **0.423** | **0.484** | **0.488** | **0.313** | **0.405** | **0.392** | **0.410** |
| Multi-output Predictor | 0.403 | 0.435 | 0.492 | 0.491 | 0.317 | 0.407 | 0.399 | 0.425 |
| Attention Layers | 0.399 | 0.452 | 0.492 | 0.508 | 0.334 | 0.407 | 0.409 | 0.451 |

# R  REPLACE MOPE WITH ALTERNATIVE DESIGNS

Here we provide the complete results of alternative designs for TFPS.

As show in Table 15, we have conducted addition experiments where we replaced the MoPE module with weighted multi-output predictor and stacked self-attention layers, keeping all other components and configurations identical. The results demonstrate that our proposed method significantly outperforms them, which validates the importance of the Top-K selection and pattern-aware design in enhancing the model's representation capacity. In contrast, multi-output predictor and self-attention typically treats all data points uniformly, which may limit its ability to capture subtle distribution shifts or evolving patterns across patches.

