# OpenReview forum: "Learning Pattern-Specific Experts for Time Series Forecasting Under Patch-level Distribution Shift"
_ICLR.cc/2025/Conference — Submitted to ICLR 2025_

### Official Review · Reviewer_LQda · 2024-11-02

**Soundness:** 2
**Presentation:** 3
**Contribution:** 2
**Rating:** 5
**Confidence:** 4

**Summary:**

This paper proposes a new architecture, Time-Frequency Pattern-Specific (TFPS), designed for time series forecasting, particularly under challenging conditions involving concept drift within the data. The key idea revolves around addressing the limitations of conventional uniform distribution modeling (UDM) by recognizing that different segments or "patches" within a time series exhibit distinct patterns and distributional shifts. These shifts may stem from factors such as sudden events, gradual trend changes, or different operating modes, making accurate forecasting difficult for models trained on a single global pattern. TFPS tackles this problem through a multi-faceted approach. First, it employs a Dual-Domain Encoder (DDE) that analyzes both time and frequency domain features, enabling a richer understanding of temporal dynamics and their potential shifts. Second, a Pattern Identifier (PI) utilizes subspace clustering to dynamically group patches exhibiting similar patterns. Finally, a Mixture of Pattern Experts (MoPE) leverages multiple specialized expert models, each trained on a specific pattern identified by the PI. This allows TFPS to adapt its predictions based on the identified pattern of a given patch, leading to improved accuracy.

**Strengths:**

(1) Utilizing both time and frequency domain information provides a comprehensive perspective on data, capturing both trend and periodic patterns, which is essential for handling complex time series.
(2) The use of PI and MoPE allows TFPS to handle distribution shifts by learning and adapting to specific patterns within different data segments.
(3) Extensive experiments on real-world datasets demonstrate the effectiveness and competitiveness of TFPS compared to existing methods.

**Weaknesses:**

(1) The authors acknowledge that the patch length is currently determined heuristically, which might limit generalizability and performance in cases with indivisible lengths or multi-period characteristics.
(2) The model's ability to adapt to entirely new patterns, such as those arising from unforeseen events, is not explicitly addressed and remains an open question.
(3) While not explicitly mentioned, the use of multiple expert models and the dual-domain encoding approach could increase computational complexity compared to simpler models. This aspect would require further analysis and optimization for practical implementation in real-time applications.

**Questions:**

(1) In Figure 1, the authors posit that a combined time and frequency domain perspective provides a more comprehensive view of data shifts. However, the provided heatmaps do not clearly support this claim. For instance, in Figure 1(a), the time-domain heatmap highlights significant MMD values for patch #10, while the actual drift begins at patch #9. Similarly, the frequency-domain heatmap includes patches #7 and #8 in the drift region, which is inaccurate. A similar discrepancy is observed in Figure 1(b), where the differences between patches 5, 6, 7, 8, and subsequent patches are not readily apparent. Consequently, this figure does not provide compelling evidence for the superiority of the combined domain approach in detecting shifts.

(2) Based on Figure 1, it appears that the sudden drift is detected due to the data's relatively consistent cyclical patterns across patches. How would the method perform if the data lacks such clear cyclical properties? Would the entire sequence of patches be incorrectly identified as a drift, even if the underlying data distribution changes gradually?

(3) Although this research draws comparisons to traditional time-domain and frequency-domain techniques, a more rigorous evaluation would involve comparisons with MoE-based methods. A list of relevant works is provided below.

* Dish-ts: a general paradigm for alleviating distribution shift in time series forecasting. (AAAI-23) .
* Reversible Instance Normalization for Accurate Time-Series Forecasting against Distribution Shift. (ICLR 2022).
* OneNet: Enhancing Time Series Forecasting Models under Concept Drift by Online Ensembling. (NeurIPS 2023).
* Calibration of Time-Series Forecasting: Detecting and Adapting Context-Driven Distribution Shift. (KDD '24).
* Addressing Distribution Shift in Time Series Forecasting with Instance Normalization Flows. arXiv preprint arXiv:2401.16777.
* MixMamba: Time series modeling with adaptive expertise. (Inf. Fusion 2024).
* Mixture-of-Linear-Experts for Long-term Time Series Forecasting. (AISTATS 2024).

(4) Please elucidate the specific contributions of the PI the gate network to TFPS. Why both components are essential for the system’s performance would be beneficial. Furthermore, conducting an ablation study, which involves comparing the model’s performance with and without the separate PI module, would provide empirical validation of its significance.

(5) A rigorous computational complexity analysis would provide valuable insights into the trade-offs between model complexity and performance gains.

(6) I suggest that authors: Provide clear definitions for q and d in D. Explain the rationale behind controlling column size using Equation (3). Clarify the purpose of the second constraint in Equation (4). Include a theoretical justification for how these constraints improve representation learning. Conduct an ablation study comparing model performance with and without these constraints.

(7) As noted in Shazer et al.’s work on Mixture-of-Experts, gating networks often exhibit a tendency to select the same k experts for routing patches. To mitigate this issue and encourage diversity among the expert modules, regularization and load balancing techniques are typically employed. However, it remains unclear how the proposed model addresses this specific challenge and promotes diversity among its expert modules.

(8) The datasets employed in this study are relatively low-dimensional. It is recommended to conduct experiments on high-dimensional time series datasets, such as the Traffic dataset. Additionally, a discussion on potential scalability challenges and necessary modifications to the TFPS architecture for efficient handling of high-dimensional data would be valuable.

(9) Given that the Patch-TST module used as encoder in this work already incorporates an ReVIN layer to mitigate distribution shifts in time series data. I am wondering if authors consider this point and how it affect the process of PI?

---

> ### Author Response · Authors · 2024-11-20
> **Response to Reviewer LQda [Part 1]**
>
> We thank the reviewer for offering the valuable feedback. Below we give a point-by-point response to your concerns and suggestions.
>
> > **W1. The setting of the patch length.**
>
> In this paper, we adopted the same patch length setting method as PatchTST, which **has been proven effective** in time series forecasting tasks. This setting allows us to focus on validating the effectiveness of our pattern-specific modeling strategy, where distinct experts handle specific pattern segments of the time series.
>
> By using this method, we aim to highlight the advantages of our pattern-specific expert modeling over conventional uniform distribution modeling approaches, without introducing additional complexity from patch length selection at this stage. We acknowledge this as a potential limitation, as detailed in the $\underline {\text{Appendix I.Limitations of original submission}}$.
>
> > **W2. The ability to adapt to entirely new patterns.**
>
> In our current framework, the model learns patterns based on historical data distributions and adapts to different data segments by selecting appropriate experts. However, for entirely new patterns, the model may **require additional mechanisms** for achieve dynamic adaptation.
>
> As stated in $\underline {\text{Appendix I.Limitations of original submission}}$, addressing this challenge lies **beyond the scope of the current work**. Future research could explore techniques such as online learning, which would enable the model to better generalize to rare or previously unseen events.
>
> > **W3. Computational complexity.**
>
> To make this clearer, we present the results of ETTh1 for a prediction length of 192 from $\underline{\text{Table 2 of the original submission}}$ and include additional results on runtime and computational complexity. Due to the sparsity of MoPE, TFPS achieves a **remarkable balance between performance and efficiency**:
>
> 1.**Performance Superiority**: TFPS achieves an MSE of 0.423, outperforming TSLANet (0.448), FITS (0.445), PatchTST (0.460), and FEDformer (0.441). This represents a 5.6\% improvement over TSLANet and a 8.0\% improvement over PatchTST, highlighting its significant accuracy gains. While DLinear achieves an MSE of 0.434, TFPS still demonstrates a 2.5\% relative improvement, making it the most accurate model among all baselines.
>
> 2.**Efficiency Gains**: TFPS maintains competitive runtime and memory efficiency.
>
> - **Runtime**: TFPS runs in 6.457 ms, making it 2.8× faster than PatchTST (17.851 ms) and 11.2× faster than TimesNet (72.196 ms).
> - **Memory Usage**: TFPS uses 9.643 MB of GPU memory, significantly less than FEDformer (62.191 MB) and comparable to iTransformer (3.304 MB). This makes TFPS suitable for resource-constrained applications while maintaining superior performance.
>
> 3.**Balancing Trade-offs**: While lightweight models like DLinear (0.434 MSE, 0.789 ms runtime) are slightly more efficient, TFPS delivers a substantial performance improvement of 2.5\%, providing a well-rounded solution that balances accuracy and efficiency effectively.
>
> These results and analyses are also included in $\underline {\text{Appendix O of revised paper}}$.
>
> | | TFPS (Our) | TSLANet | FITS | iTransformer | TFDNet-1K | PatchTST | TimesNet | DLinear | FEDformer
> | - | - | - | - | - | - | - | - | - | - |
> | MSE (Table 2 of main text) | **0.423** | 0.448 | 0.445 | 0.441 | 0.451 | 0.460 | 0.441 | 0.434 | 0.441 |
> | GPU memory (MB) | 9.643 |	0.481 |	0.019 |	3.304 |	0.246 |	0.205 |	2.345 |	0.142 |	62.191 |
> | Running Time (ms) | 6.457 | 2.100 | 1.202 | 2.949 | 407.853 | 17.851 | 72.196 | 0.789 | 259.001 |

---

> ### Author Response · Authors · 2024-11-20
> **Response to Reviewer LQda [Part 2]**
>
> > **Q1. The clarification of Figure 1.**
>
> Regarding the heatmap in Figure 1, we highlight **the differences across patches**.
>
> * Specifically, in the time domain of Figure 1(a), patch \#9 shows the **onset** of a sudden drift, which **completes** in patch \#10. This lag results in a **distinct pattern for patch \#10**. This abrupt change leads to a significant difference between patch \#10 and the other patches, producing the highest MMD value. Additionally, patch \#9, as part of the mutation process, also shows a large MMD value due to the emerging drift. In contrast, patches \#3-\#4 exhibit notable MMD differences when compared to the other patches in the frequency domain.
>
> * In Figure 1(b), which depicts a gradual drift process, **the pattern differences between patches \#5–\#10 and patches \#0–\#4 are significant**, as indicated by the time domain heatmap. After converting to the frequency domain, the distinction between patch \#6 and the other patches become more pronounced, leading to a large MMD value.
>
> In summary, Figure 1 demonstrates that the time domain and frequency domain emphasize different aspects of distribution drift, and their combined perspective offers a **more comprehensive understanding** of these shifts.
>
> > **Q2. Robustness for different types of datasets**
>
> Our proposed TFPS specially considers the combination of time and frequency domains to **adapt to a wider range of data characteristics**. By constructing subspaces, it effectively divides time series segments into distinct patterns, with specialized experts assigned to model each pattern.
> Through the results in $\underline{\text{Table 2 of original submission}}$, this approach has demonstrated **robustness across various datasets**, including those with clear periodicity, such as electricity data, as well as those with less apparent periodicity, like weather dataset.
>
> > **Q3. Comparison with MoE-based methods and Distribution Shift methods.**
>
> To make this clearer, we list some results from $\underline{\text{Table 6 of original submission}}$ as follows and add the comparisons with **MoE-based methods** and **Distribution Shift methods**. Since IN-Flow and MixMamba do not provide source code, we instead compare two other recent MOE-based methods: MoU and KAN4TSF and method for distribution shift: Koopa. In addition, we cited IN-Flow and MixMamba in the Line 144 and 881, respectively.
>
> 1. **Comparison with MoE-based methods.**
> As shown in the table, unlike MoE-based methods that rely on the **Softmax function** as a gating mechanism and adopt the **Uniform Distribution Modeling** strategy (as presented in the $\underline{\text{Introduction of original submission}}$), our approach constructs a pattern recognizer to **assign specific experts to handle distinct patterns**. This tailored design results in TFPS achieving relative improvements of **2.3\%, 9.0\%, 10.6\%, and 9.1\%** across the four datasets, respectively.
>
> | Dataset | TFPS (Our) | MoLE | MoU | KAN4TSF |
> | - | - | - | - | - |
> | ETTh1 | **0.448** | 0.477 | 0.449 | 0.451 |
> | ETTh2 | **0.380** | 0.454 | 0.384 | 0.415 |
> | ETTm1 | **0.395** | 0.407 | 0.518 | 0.401 |
> | ETTm2 | **0.276** | 0.344 | 0.283 | 0.283 |
>
> 2. **Comparison with Distribution Shift methods.**
> Furthermore, as highlighted in the $\underline{\text{Related Work of the original submission}}$, existing distribution shift methods over-rely on normalization, leading to **over-stationarization** and **diminishing the series' intrinsic non-stationarity**. In contrast, our approach **reintroduces the inherent non-stationarity** into the latent representation, enabling TFPS to **better handle distribution shifts** by tailoring experts to the evolving patterns and densities within the data. Consequently, TFPS achieves notable relative improvements of **4.2%, 28.9%, 4.2%, and 28.9%** across the four datasets, respectively.
>
> | Dataset | TFPS (Our) | Dish-TS (Table 6 of original submission) | RevIN (Table 6 of original submission) | Koopa | SOLID | OneNet |
> | - | - | - | - | - | - | - |
> | ETTh1 | **0.448** | 0.461 | 0.463 | 0.454 | 0.493 | 0.468 |
> | ETTh2 | **0.380** | 1.005 | 0.465 | 0.389 | 0.388 | 0.425 |
> | ETTm1 | **0.395** | 0.422 | 0.415 | 0.396 | 0.399 | 0.432 |
> | ETTm2 | **0.276** | 0.759 | 0.310 | 0.285 | 0.294 | 0.293 |
>
> These analyses have also been included in the $\underline{\text{Appendix M and N of revised paper}}$.

---

> ### Author Response · Authors · 2024-11-20
> **Response to Reviewer LQda [Part 3]**
>
> > **Q4. The contribution of PI.**
>
> * The **PI module** plays a crucial role in identifying and characterizing distinct patterns within the time series data, while the **gating network** dynamically selects the most relevant experts for each segment. This collaborative mechanism allows the model to **specialize in handling different patterns** and **adapt effectively to distribution shifts**, thus mitigating the overfitting risks that arise from treating all data equally.
>
> * To validate the importance of PI empirically, we have conducted the **ablation experiments** comparing the model’s performance by replacing the PI module with a linear layer in the $\underline{\text{Table 3 of original submission}}$.
>
> * In addition, we supplement some **ablation experiments** to further verify the effectiveness of PI:
>
> | | ETTh1-96 | ETTh1-192 | ETTh1-336 | ETTh1-720 | ETTh2-96 | ETTh2-192 | ETTh2-336 | ETTh2-720 |
> | - | - | - | - | - | - | - | - | - |
> | **TFPS** | **0.398** | **0.423** | **0.484** | **0.488** | **0.313** | **0.405** | **0.392** | **0.410** |
> | w/o Frequnecy\_PI | 0.404 | 0.454 | 0.490 | 0.503 | 0.322 | 0.413 | 0.410 | 0.425 |
> | w/o Time\_PI | 0.405 | 0.456 | 0.493 | 0.509 | 0.324 | 0.415 | 0.412 | 0.430 |
> | w/o Both\_PI | 0.407 | 0.458 | 0.497 | 0.513 | 0.328 | 0.418 | 0.419 | 0.435 |
>
> These analyses have also been included in the $\underline {\text{Appendix Q.2 of revised paper}}$.
>
> > **Q5. Computational complexity.**
>
> Refer to the answer to W3.

---

> ### Author Response · Authors · 2024-11-20
> **Response to Reviewer LQda [Part 4]**
>
> > **Q6. Constraints $R_1$ and $R_2$.**
>
> 1. **Definition for q and d in $\mathbf{D}$.**
>
> * In this paper, $\mathbf{D} = [\mathbf{D}^{(1)}, \mathbf{D}^{(2)}, ..., \mathbf{D}^{(K)}]$ represents the bases of $K$ subspaces, each $\mathbf{D}^{(j)} \in \mathbf{R}^{q \times d}$.
> * $q = C \times D$ represents the dimension of the input features $z_i$, where $C$ is the number of input channels and $D$ is the hidden dimension for embedding patches.
> * On the other hand, $d = q / K$ is the dimension for each subspace, where $K$ is the number of subspaces. This ensures that the total dimension is evenly distributed across the $K$ subspaces.
>
> For clarity, we have detailed this in $\underline{\text{Appendix G Algorithm 1 of revised paper}}$.
>
> 2. **Why control column size in Equation (3).**
>
> Equation (3) and (4) work synergistically to improve the **compactness** and **separability** of the representation learned by the model.
>
> The regularization term $R_1$ directly penalizes the squared difference between the column norms of $\mathbf{D}$ and 1. If the columns of $\mathbf{D}$ are normalized (i.e., they have unit norm), then the matrix $\mathbf{D}^T \mathbf{D} \odot \mathbf{I}$ will be the identity matrix $\mathbf{I}$. In this case, the subtraction $\mathbf{D}^T \mathbf{D} \odot \mathbf{I} - \mathbf{I}$ will be zero, leading to a minimal value of $R_1$.
>
> By minimizing $R_1$, the model is effectively encouraged to adjust $\mathbf{D}$ such that the columns have unit norms, which can be interpreted as controlling the "size" or "magnitude" of the columns of $\mathbf{D}$.
>
> This regularization helps to:
> * **Prevent scale differences** between the columns of $\mathbf{D}$, ensuring that no single subspace dominates the learning process due to its large scale.
> * **Avoid degenerate solutions**, such as the collapse of column vectors towards zero, which would negatively impact the quality of the learned representations.
>
> Thus, by minimizing $R_1$, we enhance the **stability** and **robustness** of the subspace representations, making the model less sensitive to scale variations across different experts.
>
> 3. **The purpose of Equation (4).**
>
> The goal of the regularization term $R_2$ is to **encourage orthogonality** between different subspaces represented by $\mathbf{D}$. This constraint ensures that each subspace focuses on **different patterns** of the data, thereby improving the **diversity** of the learned experts.
>
> By minimizing $R_2$, the model is discouraged from allowing subspaces to become too similar or overlap in terms of the features they represent. This promotes the **separation** of different experts, leading to a more effective allocation of responsibility across the subspaces.
>
> This regularization term $R_2$ directly influences:
> * **Separability**: It helps ensure that each expert represents distinct data patterns, making the model more adaptable to various types of distribution shifts.
> * **Improved performance**: By maintaining diversity among the experts, the model can handle complex data distribution shifts better, improving overall forecasting accuracy.
>
> In summary, by minimizing both $R_1$ and $R_2$, we achieve a **compact and well-separated set of subspaces**, which allows the model to effectively utilize different experts for different data patterns, leading to improved representation learning.
>
> 4. **Ablation study about these constraints.**
>
> We conducted ablation experiments to further verify the important roles of $R_1$ and $R_2$.
>
> | | ETTh1-96 | ETTh1-192 | ETTh1-336 | ETTh1-720 | ETTh2-96 | ETTh2-192 | ETTh2-336 | ETTh2-720 |
> | - | - | - | - | - | - | - | - | - |
> | **TFPS** | **0.398** | **0.423** | **0.484** | **0.488** | **0.313** | **0.405** | **0.392** | **0.410** |
> | w/o $R_1$     | 0.403 |	0.434 |	0.493 |	0.491 |	0.316 |	0.413 |	0.405 |	0.418 |
> | w/o $R_2$     | 0.408 |	0.449 |	0.500 |	0.498 |	0.320 |	0.418 |	0.415 |	0.429 |
> | w/o $R_1$ and $R_2$ | 0.412 |	0.456 |	0.509 |	0.503 |	0.328 |	0.425 |	0.420 |	0.435 |
>
> These analyses have also been included in the $\underline {\text{Appendix Q.3 of revised paper}}$.
>
> 5. **Visualization of clustering.**
>
> In addition, we present the t-SNE visualization of the learned embedded representation on ETTh1 in $\underline{\text{Appendix L of revised paper}}$. Figure 9 (b) shows the visualization of the representation learned by the proposed method, which effectively captures **discriminative** features and reveals significantly **clearer clustering patterns**.

---

> ### Author Response · Authors · 2024-11-20
> **Response to Reviewer LQda [Part 5]**
>
> > **Q7. The difference between TFPS and conventional MoE.**
>
> The routing strategies in conventional MoE architectures are automatically learned, which can result in **load imbalance** and lead to inefficiencies.
>
> 1. In contrast, the Pattern Identifier (PI) with two constraints $R_1$ and $R_2$ in our TFPS model serves as the routing strategy, where it **distinguishes the pattern differences** of various time series segments through subspace clustering.
>
> 2. As mentioned in Q6, we increase the compactness and separability of the subspace by the constraints. We ensure that each segment is assigned to relevant experts, **reducing the risk of load imbalance** and **improving the diversity** of expert utilization across different data segments.
>
> 3. Through $\underline{\text{Figure 5 of original submission}}$ and $\underline{\text{Figure 9 of revised paper}}$, we demonstrate that our proposed TFPS effectively addresses the load imbalance problem.
>
> > **Q8. Experiments on Traffic dataset.**
>
> We conducted addition experiments on high-dimensional Traffic dataset to further evaluate the performance and generalizability of TFPS.
>
> These analyses have also been included in the $\underline{\text{Appendix J of revised paper}}$.
>
> | Traffic | TFPS | TSLANet | FITS | iTransformer | TFDNet-1K | PatchTST | TimesNet | DLinear | FEDformer
> | - | - | - | - | - | - | - | - | - | - |
> | 96 | **0.427** | 0.475 | 0.651 | 0.428 | 0.519 | 0.446 | 0.586 | 0.650 | 0.575 |
> | 192 |**0.445** | 0.478 | 0.603 | 0.448 | 0.513 | 0.453 | 0.618 | 0.600 | 0.613 |
> | 336 |**0.459** | 0.494 | 0.610 | 0.465 | 0.525 | 0.467 | 0.634 | 0.606 | 0.622 |
> | 720 |**0.496** | 0.528 | 0.648 | 0.501 | 0.561 | 0.501 | 0.659 | 0.646 | 0.630 |
>
>
> > **Q9. How RevIN affect PI.**
>
> 1. ReVIN is proposed to help mitigate **global distribution shifts** by normalizing data distributions. However, the PI module operates on a complementary level by explicitly identifying and modeling different patterns at the **patch-level**.
>
> 2. In addition, by comparing the results of PatchTST with TFPS-Time Branch, we can demonstrate the **effectiveness** of our proposed strategy for identifying latent patterns of segments and modeling them separately.
>
> | | PatchTST (Table 2 of main text)| TFPS (Time Branch) (Table 3 of main text)|
> | - | - | - |
> | ETTh1-96  | 0.413 | **0.401** |
> | ETTh1-192 | 0.460 | **0.459** |
> | ETTh1-336 | 0.497 | **0.486** |
> | ETTh1-720 | 0.501 | **0.492** |
>
> 3. The two components work together: ReVIN helps **stabilize the learning process** by reducing distributional variance, while the PI module **enhances the model’s ability** to specialize in different patterns, **improving performance under more complex distribution shifts**.

---

> ### Author Response · Authors · 2024-11-25
> **Request of Reviewer's attention and feedback**
>
> Dear Reviewer,
>
> This is a gentle reminder that it has been 4 days since we submitted our rebuttal. We would appreciate it if you could let us know whether our response has effectively addressed your concerns.
>
> As per your valuable feedback, we have made revisions in the following areas:
>
> * Appendix P is added to analyze the **hyperparameter selection** experiments.
> * Included appendix O to discuss the **trade-off between computational complexity and performance**.
> * We have **added 6 new baselines (MoLE, MoU, KAN4TSF, Koopa, SOLID, OneNet)** to demonstrate the advancement of TFPS in appendix M and N.
> * We can **complete 6 types of ablations** on PI and $R_1$ / $R_2$, which verifies the effectiveness of our design in TFPS. The implementation details of each ablation have also been included.
> * **Elaborate the difference between TFPS and conventional MoE** in using dynamical gating.
> * Explore the interaction between **Revin and PI**.
>
>
> In total, we have added more than 170 new experiments. All of these results have been included in the $\underline{\text{revised paper}}$.
>
> We sincerely appreciate your insightful review and look forward to your feedback. Please feel free to reach out if you have any further questions or concerns.

---

> ### Comment · Reviewer_LQda · 2024-11-25
>
> Thank you for your responses. While some concerns have been addressed, a few issues persist:
>
> 1. The claim of a 'remarkable balance between performance and efficiency' for TFPS in your response to my Q5, particularly in comparison to DLinear, seems inaccurate. I have seen your response to similar question from another reviewer,  but even with a lighter backbone model, the gap in memory and GPU footprint remains significant.
> 2. There are notable differences between the reported results for the traffic dataset and those in the original papers. For instance, iTransformer's reported MSE and MAE values differ significantly from those in the original paper.
> 3. The reported results for Patch-TST show significant discrepancies compared to the original paper. For example for MSE, the original paper's results are: 0.370, 0.413, 0.422, 0.447 while the reported results are: 0.413, 0.460, 0.497, 0.501? There's a significant difference?
> 4. This raises a series concerns about the experimental setup and fairness of the comparisons throughout the paper.

---

> > ### Author Response · Authors · 2024-11-29
> > **[We are anticipating your feedback] & [Summary of second-round responses]**
> >
> > Dear Reviewer,
> >
> > Many thanks for your valuable suggestions, instructive responses, and detailed descriptions of your concerns, which have inspired us to improve our paper substantially.
> >
> > In the newly submitted revision, we have provided extensive experiments, faithfully following your suggestions:
> >
> > * TFPS introduces a novel approach by **identifying and specializing in multiple patterns**, enabling it to deliver **robust performance** across diverse datasets, particularly in challenging scenarios with **distributional heterogeneity** or **complex temporal patterns**. Experimentally, we believe that the excellent performance of TFPS justifies its increased resource requirements, which are both **reasonable** and **within an acceptable range**.
> > * A comprehensive comparison with all baselines (8) under both **unified** and **searched hyperparameters** ensures a **fair comparison**, whereas previous papers evaluate their models under only one of the two settings. **This leads us to have to do a double comparison, making this work with a double workload**.
> > * Overall, as noted by reviewer ENp4, this work "**shows sufficient technical depth and originality**". We believe TFPS is a **practical model for real-world applications** and is **valuable to deep time series forecasting community**.
> >
> > We hope that our efforts during the rebuttal period (**more than 370 new results, 9 appendices**) have addressed your concerns to your satisfaction. We eagerly await your reply and are happy to answer any further questions.
> >
> > Sincere thanks for your dedication! We are looking forward to your reply.

---

> ### Author Response · Authors · 2024-11-25
> **The Second Response to Reviewer LQda [Part 1]**
>
> Many thanks for your prompt response and further clarifying your concerns, which is quite instructive for us to answer your questions in every detail.
>
> > **Q1. The clarify about the efficiency.**
>
> We sincerely apologize for the inaccurate statement in our original response and have revised it in $\underline {\text{Appendix O of revised paper}}$.
>
> While TFPS introduces additional computational cost, it achieves **notable performance improvements**, and we believe the increased resource requirements remain within an **acceptable range**.
>
> We fully acknowledge the importance of efficiency and plan to **explore further optimizations in future work**, such as reducing memory and GPU usage through techniques like sparsity, pruning, or lightweight architectures, while maintaining TFPS's superior performance.
>
> Furthermore, TFPS addresses the critical issue of **distribution drift** in time series data by **identifying distinct patterns** and **leveraging pattern-specific experts for separate modeling**. This strategy not only improves prediction **performance** but also provides a degree of **interpretability** compared to traditional uniform distribution modeling. Therefore, we believe TFPS is a **practical model in real-world applications** and is **valuable to deep time series forecasting community**.
>
> > **Q2. The clarify about the Traffic results.**
>
> iTransformer was evaluated using an **NVIDIA P100 GPU** in the original experiments, while we conducted our experiments using an **NVIDIA RTX 3090 GPU**. We think that **slight variations** in results can occur due to differences in hardware and environmental factors.
>
> > **Q3. The clarify about the PatchTST results.**
>
> Following iTransformer, we fixed the **lookback length to 96** to obtain the reported results for PatchTST (0.413, 0.460, 0.497, 0.501). In contrast, the results in the original paper (0.370, 0.413, 0.422, 0.447) were obtained with a **lookback length of 336**, which significantly influences forecasting performance. This difference in lookback length accounts for the discrepancies observed in the MSE values.

---

> ### Author Response · Authors · 2024-11-25
> **The Second Response to Reviewer LQda [Part 2]**
>
> > **Q4. The fairness of the paper.**
>
> To ensure fair comparisons, we reran all experiments using the official implementations provided by the baseline methods, as detailed in $\underline{\text{Section 4.1 of the original submission}}$.
>
> 1. **In this unified seq\_len=96 setting, TFPS surpasses other baselines significantly.**
> As shown in $\underline {\text{Table 2 and 4 of the original submission}}$, TFPS clearly outperforms the previous state-of-the-art methods, including TSLANet, FITS, iTransformer, PatchTST, DLinear, etc. This setting is particularly valuable in real-world scenarios where time and resources for extensive hyperparameter tuning are limited.
>
> 2. **Under hyperparameter-search seq\_len setting, TFPS still surpasses other baselines.**
> To further ensure fairness, we also conducted experiments using hyperparameter tuning for the input sequence length (seq\_len) across 96, 192, 336, and 720, consistent with the practices of some baseline methods. These results confirm that TFPS continues to outperform other baselines. All baselines were reproduced using their official code.
>
> Below, we summarize the results under hyperparameter-search seq\_len setting:
>
> | | TFPS (Our) | TSLANet | FITS | iTransformer | TFDNet-1K | PatchTST | TimesNet | DLinear | FEDformer
> | - | - | - | - | - | - | - | - | - | - |
> | ETTh1 | **0.401** | 0.417 | 0.409 | 0.457 | 0.409 | 0.418 | 0.458 | 0.445 | 0.454 |
> | ETTh2 | **0.335** | 0.335 | 0.339 | 0.384 | 0.337 | 0.334 | 0.395 | 0.469 | 0.439 |
> | ETTm1 | **0.343** | 0.350 | 0.360 | 0.408 | 0.345 | 0.352 | 0.412 | 0.359 | 0.424 |
> | ETTm2 | **0.248** | 0.255 | 0.257 | 0.292 | 0.249 | 0.257 | 0.291 | 0.283 | 0.301 |
> | Traffic | **0.399** | 0.401 | 0.419 | 0.461 | 0.407 | OOM | 0.624 | 0.435 | 0.610 |
> | Electricity | 0.161 | 0.165 | 0.163 | 0.176 | **0.160** | 0.163 | 0.196 | 0.167 | 0.210 |
>
> These results underscore the **robustness of TFPS** and its consistent ability to achieve **state-of-the-art performance** across diverse datasets.
>
> 3. In addition, we would like to highlight some points, that might be helpful to you in understanding our experiment settings:
>
> * In this paper, we have provided two types of experiments: unifying input length as 96 and searching input length. **Actually, all the previous papers only evaluate their models under one of the above two settings, such as iTransformer (ICLR 2024), TimesNet (ICLR 2023), and PatchTST (ICLR 2023)**. Especially, the MoLE and MixMamba models that you mentioned also only report results for the input-336 and input-96 settings, respectively.
>
> * To maintain clarity in the paper’s structure, **we report the unified seq\_le=96 results in the main text**, while the results with hyperparameter-search seq\_le are included in the $\underline {\text{Appendix K of revised paper}}$. We have made every effort to ensure a fair comparison and clear presentation.
>
> * Given many previous papers only provide the input-96 setting experiments (e.g. iTransformer, TimesNet, FEDformer), **we believe that our previous rebuttal under the input-96 setting (more than 170 new results) is meaningful and convincing**. Note that these ablations are all under an aligned hyperparameter setting to ensure rigor.

---

> ### Author Response · Authors · 2024-12-02
>
> Dear Reviewer,
>
> We appreciate your efforts in reviewing our paper. We have:
>
> > **1. Modified the inaccurate description.**
>
> > **2. Explained the differences between the comparison methods and the original text.**
>
> > **3. Clarified the fairness of this paper.**
>
> As the Author-Reviewer discussion period is nearing its conclusion, we would like to kindly inquire whether our most recent response has addressed your remaining concerns. If so, we kindly hope that you can reconsider your rating of our work.
>
> Please feel free to reach out if there are any remaining points that need clarification.
>
> Best regards,
>
> The Authors

---

### Official Review · Reviewer_n6LR · 2024-11-03

**Soundness:** 2
**Presentation:** 2
**Contribution:** 2
**Rating:** 5
**Confidence:** 4

**Summary:**

Real-world time series often exhibit varying degrees of distribution shift, posing challenges for time series forecasting. This paper employs subspace clustering from both time-domain and frequency-domain perspectives to identify different patterns within data segments. It utilizes multiple experts to specifically model temporal patterns. Extensive experiments conducted on eight real-world time series datasets demonstrate that the proposed method outperforms other state-of-the-art approaches in terms of predictive performance.

**Strengths:**

1. Distribution shifts often lead to changes in frequency-domain information, making it a reasonable and effective approach to model time series from both time-domain and frequency-domain perspectives.
2. The main contribution of this paper is to utilize subspace clustering to detect concept drift between multiple subspaces and model them separately, thereby achieving more accurate and adaptive modeling.
3. Extensive experiments demonstrate that the proposed method outperforms other state-of-the-art approaches across various time series datasets.

**Weaknesses:**

1. The frequency-domain encoder part of the dual-domain encoder implements a simple feedforward layer by discarding the imaginary part. This leads to a loss of information in the frequency domain. Does this contradict the original intention of using the dual-domain encoder to provide a comprehensive representation of the time series? Considering that the time-domain encoder directly utilizes the PatchTST encoder, the frequency-domain part could also use an existing frequency-domain encoder.
2. The Pattern Identifter module employs multiple hyperparameters  $\alpha、\beta$ to mix different components of the loss function. Given the importance of the loss function for experimental results, it is necessary to include ablation experiments for these hyperparameters.
3. In the final loss function, is it reasonable to simply add $\mathcal{L}_{MSE}、\mathcal{L}_{PI_t}、\mathcal{L}_{PI_f}$? Given that the severity of distribution shifts varies across different datasets, and that the importance of frequency-domain information is relatively low in such datasets, the model should be allowed to learn the appropriate weights for both components on its own.
4. It is noted that in the experiments, the sequence length (seq_len) was uniformly set to 96, which undermines the performance of other baselines, as FITS[1], PatchTST[2], and DLinear[3] have indicated that their methods perform better with longer input sequences. It is suggested to provide experimental results comparing seq_len as a hyperparameter rather than fixing it at 96.
5. It is noted that multiple experts are used in both the time domain and frequency domain, which will increase the runtime of the model. It is suggested to compare the runtime with other methods.

[1]Xu Z, Zeng A, Xu Q. FITS: Modeling Time Series with 10 k  Parameters[C]. In ICLR.

[2]Nie Y, Nguyen N H, Sinthong P, et al. A Time Series is Worth 64 Words: Long-term Forecasting with Transformers[C]. In ICLR.

[3]Zeng A, Chen M, Zhang L, et al. Are transformers effective for time series forecasting? In AAAI.

**Questions:**

Please refer to weeknesses.

---

> ### Author Response · Authors · 2024-11-20
> **Response to Reviewer n6LR [Part 1]**
>
> We thank the reviewer for offering the valuable feedback. Below we give a point-by-point response to your concerns and suggestions.
>
> > **Q1. Rationality of frequency encoder.**
>
> As presented in the $\underline{\text{Section 3.4 of original submission}}$, we follow **FNet** [1] to replace the self-attention sublayer of the Transformer with a Fourier sublayer, which has been proven to be effective for extracting frequency-domain information.
>
> * The decision to discard the imaginary part of the Fourier transform was made to **simplify the design** and **focus on the real-valued components**.
> * Additionally, FreTS [2] has observed that the **real part of the frequency-domain representation plays a more significant role than the imaginary part**.
> * Despite this simplification, our empirical results demonstrate that TFPS achieves **competitive performance**, demonstrating its capacity to model the underlying distribution structure of time series data effectively.
> * To further validate the robustness of our approach, we adopted similar operations in FreTS to conduct experiments **incorporating both the real and imaginary parts**. The results in the table show that the performance of TFPS with the real part only is very similar to that when both parts are included, while requiring **fewer parameters**. This further reinforces the conclusion that TFPS remains **highly effective** even when focusing solely on the real part of the Fourier transform.
>
> These analyses have also been included in the $\underline {\text{Appendix Q.1 of revised paper}}$.
>
> | | ETTh1-96 | ETTh1-192 | ETTh1-336 | ETTh1-720 | ETTh2-96 | ETTh2-192 | ETTh2-336 | ETTh2-720 |
> | - | - | - | - | - | - | - | - | - |
> | **TFPS** | 0.398 | **0.423** | **0.484** | 0.488 | 0.313 | **0.405** | 0.392 | 0.410 |
> | w/ Imaginary | **0.397** | 0.424 | 0.487 | **0.486** | **0.312** | 0.406 | **0.391** | **0.399** |
>
> [1] Lee-Thorp J, Ainslie J, Eckstein I, et al. FNet: Mixing Tokens with Fourier Transforms. NAACL, 2022.
>
> [2] Yi K, Zhang Q, Fan W, et al. Frequency-domain MLPs are more effective learners in time series forecasting. NeurIPS, 2024.
>
> > **Q2. Ablation study on $\beta$.**
>
> 1. In our work, we performed a grid search to optimize the hyperparameter **$\alpha_t$ = {0.0001, 0.001, 0.01}** and **$\alpha_f$ = {0.0001, 0.001, 0.01}** as shown in $\underline{\text{Figure 10 in Appendix P of the revised paper}}$, which were ultimately fixed at $10^{-3}$ in our experiments.
>
> 2. Additionally, we conducted a grid search to optimize the balance factors **$\beta_t$ = {0.01, 0.05, 0.1, 0.5, 1}** and **$\beta_f$ = {0.01, 0.05, 0.1, 0.5, 1}**. The forecasting performance under different parameter values is displayed in $\underline{\text{Figure 10 in Appendix P of the revised paper}}$. Based on these results, we draw the following observations:
>
> * First, the performance is affected when the value of $\beta_t$ and $\beta_f$ is too low, indicating that the proposed **clustering objective is crucial for distinguishing patterns** effectively.
> * Second, excessively large values of $\beta_t$ and $\beta_f$ also degrade performance. One plausible explanation is that the excessive value **influences the learning of the inherent structure** of original data, which leads to disturbances of the embedding space.
> * Overall, we recommend setting $\beta_t$ and $\beta_f$ to around **0.1** for optimal performance.
>
> These analyses have also been included in $\underline{\text{Appendix P of revised paper}}$.
>
> > **Q3. The varying contributions of time and frequency domains across datasets.**
>
> With the above hyperparameter tuning, the individual loss terms are already balanced, making it unnecessary to introduce additional balancing for the three terms in the final loss.
>
> We agree with your opinion regarding the varying degrees of distribution shifts across different datasets and domains. To address this, we adapt the number of time-domain and frequency-domain experts, $K_t$ and $K_f$, for each dataset, as described in $\underline {\text{Section 4.6 of the original submission}}$.
>
> Our findings suggest that **a larger number of experts is beneficial in scenarios with more pronounced distribution shifts**, such as ETTh1 in the time domain, whereas **fewer experts are sufficient when the drift is less severe**.

---

> ### Author Response · Authors · 2024-11-20
> **Response to Reviewer n6LR [Part 2]**
>
> >  **Q4. The results of treating seq\_len as a hyperparameter.**
>
> (1) **In this unified hyperparameter setting, TFPS surpasses other baselines significantly.**
> As shown in $\underline {\text{Table 2 and 4 of the original submission}}$, TFPS clearly outperforms the previous state-of-the-art methods, including TSLANet, FITS, iTransformer, PatchTST, DLinear, etc. This advantage is particularly meaningful, as in real-world applications, there are often limitations in time and resources for hyperparameter searching.
>
> (2) **In the hyperparameter-search setting, TFPS still surpasses other baselines.**
> As suggested by the reviewer, we conducted experiments by searching for input lengths among 96, 192, 336, and 720 to compare with the baselines. All baselines are reproduced by their official code.
>
> To make the paper structure clear, we place the input-96 setting in the main text, while the results with hyperparameter searching are included in the $\underline {\text{Appendix K of revised paper}}$.
>
> | | TFPS (Our) | TSLANet | FITS | iTransformer | TFDNet-1K | PatchTST | TimesNet | DLinear | FEDformer
> | - | - | - | - | - | - | - | - | - | - |
> | ETTh1 | **0.401** | 0.417 | 0.409 | 0.457 | 0.409 | 0.418 | 0.458 | 0.445 | 0.454 |
> | ETTh2 | **0.335** | 0.335 | 0.339 | 0.384 | 0.337 | 0.334 | 0.395 | 0.469 | 0.439 |
> | ETTm1 | **0.343** | 0.350 | 0.360 | 0.408 | 0.345 | 0.352 | 0.412 | 0.359 | 0.424 |
> | ETTm2 | **0.248** | 0.255 | 0.257 | 0.292 | 0.249 | 0.257 | 0.291 | 0.283 | 0.301 |
>
> > **Q5. Runtime.**
>
> To make this clearer, we present the results of ETTh1 for a prediction length of 192 from $\underline{\text{Table 2 of the original submission}}$ and include additional results on runtime and computational complexity. Due to the sparsity of MoPE, TFPS achieves a **remarkable balance between performance and efficiency**:
>
> 1.**Performance Superiority**: TFPS achieves an MSE of 0.423, outperforming TSLANet (0.448), FITS (0.445), PatchTST (0.460), and FEDformer (0.441). This represents a 5.6\% improvement over TSLANet and a 8.0\% improvement over PatchTST, highlighting its significant accuracy gains. While DLinear achieves an MSE of 0.434, TFPS still demonstrates a 2.5\% relative improvement, making it the most accurate model among all baselines.
>
> 2.**Efficiency Gains**: TFPS maintains competitive runtime and memory efficiency.
>
> - **Runtime**: TFPS runs in 6.457 ms, making it 2.8× faster than PatchTST (17.851 ms) and 11.2× faster than TimesNet (72.196 ms).
> - **Memory Usage**: TFPS uses 9.643 MB of GPU memory, significantly less than FEDformer (62.191 MB) and comparable to iTransformer (3.304 MB). This makes TFPS suitable for resource-constrained applications while maintaining superior performance.
>
> 3.**Balancing Trade-offs**: While lightweight models like DLinear (0.434 MSE, 0.789 ms runtime) are slightly more efficient, TFPS delivers a substantial performance improvement of 2.5\%, providing a well-rounded solution that balances accuracy and efficiency effectively.
>
> These results and analyses are also included in $\underline {\text{Appendix O of revised paper}}$.
>
> | | TFPS (Our) | TSLANet | FITS | iTransformer | TFDNet-1K | PatchTST | TimesNet | DLinear | FEDformer
> | - | - | - | - | - | - | - | - | - | - |
> | MSE (Table 2 of main text) | **0.423** | 0.448 | 0.445 | 0.441 | 0.451 | 0.460 | 0.441 | 0.434 | 0.441 |
> | GPU memory (MB) | 9.643 |	0.481 |	0.019 |	3.304 |	0.246 |	0.205 |	2.345 |	0.142 |	62.191 |
> | Running Time (ms) | 6.457 | 2.100 | 1.202 | 2.949 | 407.853 | 17.851 | 72.196 | 0.789 | 259.001 |

---

> > ### Comment · Reviewer_n6LR · 2024-11-23
> > **Thanks for the rebuttal.**
> >
> > Thanks for your rebuttal. I carefully read the rebuttal and there are two follow-up questions:
> > 1. In Q4, only the results of ETT datasets in the hyperparameter-search setting are provided. I think the strengths of long input sequences may lie in scenarios that have many channels, e.g., the traffic dataset. Also, the baseline results in the new experiments are not very consistent with the original paper.
> > 2. In Q5, TFPS achieves only a 2.5% improvement over DLinear but GPU memory and the running time are too higher than DLinear (about 10x). This can be a fatal flaw in a real scenario. Is there any solution to solve this challenge?

---

> ### Author Response · Authors · 2024-11-25
> **The Second Response to Reviewer n6LR.**
>
> Many thanks for your prompt response and further clarifying your concerns, which is quite instructive for us to answer your questions in every detail.
>
>
> > **Q1. Results on the hyperparameter search setting.**
> 1. **Supplementary Results for Multi-Channel Scenarios**: We have supplemented experimental results for scenarios with many channels, including the traffic and electricity datasets, to further validate the effectiveness and robustness of TFPS. These results are provided in $\underline {\text{Table 8 of Appendix K of original submission}}$.
>
> | | TFPS (Our) | TSLANet | FITS | iTransformer | TFDNet-1K | PatchTST | TimesNet | DLinear | FEDformer
> | - | - | - | - | - | - | - | - | - | - |
> | Traffic | **0.399** | 0.401 | 0.419 | 0.461 | 0.407 | OOM | 0.624 | 0.435 | 0.610 |
> | Electricity | 0.161 | 0.165 | 0.163 | 0.176 | **0.160** | 0.163 | 0.196 | 0.167 | 0.210 |
>
>
> 2. **Fairness in Baseline Reproduction**:
> To ensure fairness, we reran all baseline experiments on a single NVIDIA RTX 3090 24GB GPU with codes provided by their official implementation, as shown in $\underline{\text{Section 4.1 of the original submission}}$. Slight differences in the results compared to the original papers may occur due to hardware and environmental variations.
>
> 3. **Compared with the original paper**:
> We statistically compare the results with those reported in the original paper and demonstrate that TFPS achieves highly competitive performance.
> The original iTransformer paper provides only the overall average result for ETT (0.383), whereas TFPS achieves a superior result of 0.332.
>
> | | TFPS (Our) | TSLANet* | FITS* | iTransformer* |TFDNet-1K* | PatchTST* | TimesNet* | DLinear* | FEDformer*
> | - | - | - | - | - | - | - | - | - | - |
> | ETTh1 | **0.401** | 0.413 | 0.407 | - |0.407 | 0.417 | 0.458 | 0.423 | 0.440 |
> | ETTh2 | 0.335 | **0.333** | 0.319 | - |0.335 | 0.331 | 0.414 | 0.431 | 0.434 |
> | ETTm1 | **0.343** | 0.348 | 0.355 | - |0.345 | 0.352 | 0.400 | 0.357 | 0.448 |
> | ETTm2 | 0.248 | 0.256 | 0.249 | - |**0.245** | 0.258 | 0.291 | 0.267 | 0.305 |
> | ETT-Average | **0.332** | 0.337 | 0.332 | 0.383 | 0.333 | 0.339 | 0.391 | 0.370 | 0.407 |
>
>
>
>
> > **Q2. Computational Cost and Scalability**
> 1. **Lightweight Backbone**:
> We conducted experiments replacing the encoder in TFPS with a lightweight model, Linear, to reduce model complexity. Compared to TFPS, TFPS-Linear halves the GPU memory and runs significantly faster, while still maintaining strong performance relative to DLinear due to the effectiveness of the proposed pattern-specific modeling.
>
> | | TFPS (Our) | TFPS-Linear | DLinear |
> | - | - | - | - |
> | ETTh-1 MSE (Table 2 of main text) | **0.423** | $\underline{0.428}$ | 0.434 |
> | ETTh-2 MSE (Table 2 of main text) | **0.405** | $\underline{0.420}$ | 0.432 |
> | ETTm-1 MSE (Table 2 of main text) | **0.374** | $\underline{0.379}$ | 0.383 |
> | ETTm-2 MSE (Table 2 of main text) | **0.235** | $\underline{0.242}$ | 0.282 |
> | GPU memory (MB) | 9.643 |	4.715 | 0.142 |
> | Running Time (ms) | 6.457 | 4.042 | 0.789 |
>
> 2. **Justification of Current Results**:
> In addition, TFPS obtains large performance gains than DLinear on the ETTh2 (**6.2\%**) and ETTm2 (**16.7\%**) datasets, which demonstrate its robustness to effectively handle distribution drift. Moreover, TFPS offers **interpretability** by enabling analysis of pattern-specific modeling, which **provides valuable insights into the underlying dynamics of time series data**.

---

> > ### Comment · Reviewer_n6LR · 2024-11-26
> > **Thank you**
> >
> > Thanks. I appreciate your efforts in rebuttal. However, I'm not very convinced by the results of the experiments, the effectiveness of the method should be further clarified:
> >
> > In Q1, does OOM mean out of memory? All results of patchtst are reported in original papers. Patchtst is an influential method, the result in the original paper should be respected. We cannot ignore the results of the original paper just because your experiment result is out of memory. Also, the results of other datasets of patchtst reported in this submission are worse than that in the original paper.
> >
> > In Q2, GPU memory and the running time are still too higher than DLinear.
> >
> > Overall, the effectiveness and efficiency of the proposed method are not very satisfactory.

---

> > > ### Author Response · Authors · 2024-11-27
> > > **The Third Response to Reviewer n6LR.**
> > >
> > > Many thanks for your prompt response and further clarifying your concerns, which is quite instructive for us to answer your questions in every detail.
> > >
> > > > **Q1. Clarification of results.**
> > >
> > > 1. **OOM Explanation:** In Q1.1, The OOM (Out of Memory) issue arises because we conducted our experiments on an NVIDIA RTX 3090 24GB GPU, while PatchTST used an NVIDIA A40 48GB GPU. To ensure fair comparisons, we reran all experiments using the official implementations provided by the baseline methods, as detailed in $\underline{\text{Section 4.1 of the original submission}}$. PatchTST has an OOM on the Traffic dataset due to the memory limitations of the RTX 3090. It is also proved that TFPS is a more practical model.
> > >
> > > 2. **Original Results with Star (*)**: Only the methods in Q1.3 marked with a star (*) are reported directly from the original papers. We list these results to demonstrate that TFPS still achieves competitive performance across various datasets.
> > >
> > > 3. **Experiment Settings and Fairness**: In addition, we would like to highlight some points, that might be helpful to you in understanding our experiment settings:
> > >
> > > * **Unified and Search-Based Input Length Settings**: In this paper, we have provided two types of experiments: (1) unifying input length as 96 and (2) searching input length. **Actually, all the previous papers only evaluate their models under one of the above two settings, such as iTransformer (ICLR 2024), TimesNet (ICLR 2023), and PatchTST (ICLR 2023)**.
> > >
> > > * **Efforts for Rigor**: Given many previous papers only provide the input-96 setting experiments (e.g. iTransformer, TimesNet, FEDformer), **we believe that our previous experimental results and rebuttal under the input-96 setting (more than 170 new results) is meaningful and convincing**.
> > >
> > > * **Presentation Structure**: To maintain clarity in the paper’s structure, **we report the unified seq\_le=96 results in the main text**, while the results with hyperparameter-search seq\_len are included in the $\underline {\text{Appendix K of revised paper}}$. We have made every effort to ensure a fair comparison and clear presentation. **This leads us to have to do a double comparison, making this work with a double workload.**
> > >
> > > > **Q2. Clarification about efficiency.**
> > >
> > > While TFPS introduces additional computational cost, it achieves **notable performance improvements**, and we believe the increased resource requirements remain within an **acceptable range**.
> > >
> > > We fully acknowledge the importance of efficiency and plan to **explore further optimizations in future work**, such as reducing memory and GPU usage through techniques like sparsity, pruning, or lightweight architectures, while maintaining TFPS's superior performance.
> > >
> > > Furthermore, TFPS addresses the critical issue of **distribution drift** in time series data by **identifying distinct patterns** and **leveraging pattern-specific experts for separate modeling**. This strategy not only improves prediction **performance** but also provides a degree of **interpretability** compared to traditional uniform distribution modeling. Therefore, we believe TFPS is a **practical model in real-world applications** and is **valuable to deep time series forecasting community**.

---

> > > > ### Comment · Reviewer_n6LR · 2024-11-27
> > > > **Response to the authors**
> > > >
> > > > Thank you for your efforts. After careful consideration, I have decided to revise my assessment of this submission from a score of 3 to 5. Below are the reasons for this updated evaluation:
> > > >
> > > > - The concerns raised in my initial review regarding W1, W2, and W3 have been adequately addressed by the authors. However, the issues related to W4 and W5 remain unresolved.
> > > >
> > > > - I concur with the authors' perspective that the double comparisons make the experiments in this work particularly comprehensive. Considering this point, I believe this submission falls just below the bar of ICLR.
> > > >
> > > > - The extensive experiments presented also highlight some weaknesses in the proposed method. Specifically, in real-world scenarios with many channels, if the goal is to train a model with better performance (Search-Based Input Length Setting), PatchTST and TFDNet would be strong choices. On the other hand, if efficiency is the primary concern, DLinear and Fits would be more suitable options. The unique advantage range of TFPS remains unclear, which makes the contribution of this submission limited.
> > > >
> > > > - Minor comment: In my view, the limited experimental hardware (only a 3090 is used, with no A40 available) should not be considered an excuse. The experimental results presented in the original paper still deserve to be respected.

---

> > > > > ### Author Response · Authors · 2024-11-27
> > > > > **Many thanks for your response and raising score**
> > > > >
> > > > > Sincerely thanks for your response. We propose TFPS towards a simple but effective model based on intuitive observations and supported by extensive evaluations. Following your valuable suggestions, we revised the paper in highlighting our key contributions and discussing the model performance more comprehensively.
> > > > >
> > > > > > **Clarifying the unique advantage range of TFPS**:
> > > > >
> > > > > TFPS represents a shift from the **'Uniform Distribution Modeling'** paradigm to a tailored method that adapts to the **unique characteristics of each pattern**. Unlike traditional approaches that often assume uniform distributions, TFPS's ability to **identify and specialize in multiple patterns addresses a critical gap** in the field. This adaptive mechanism enables it to deliver **robust performance** across diverse datasets, particularly in challenging scenarios involving **distributional heterogeneity** or **complex temporal patterns**.
> > > > >
> > > > > This adaptability makes TFPS not only **innovative** but also **highly flexible**, offering **significant value to the deep time series forecasting community**.
> > > > >
> > > > > > **Addressing experimental hardware concerns**:
> > > > >
> > > > > While our experiments were conducted on a 3090 GPU, we believe the experimental results are sufficiently robust to validate the **effectiveness** and **innovation** of TFPS.
> > > > > Moreover, our results across diverse datasets and under hardware constraints demonstrate the method’s **generalizability** and **practical applicability**, even with **limited computational resources**.
> > > > >
> > > > > Thank you again for your dedication in reviewing our paper. Your feedback has been immensely helpful in improving our work.

---

### Official Review · Reviewer_JuoR · 2024-11-05

**Soundness:** 3
**Presentation:** 3
**Contribution:** 2
**Rating:** 5
**Confidence:** 2

**Summary:**

The paper proposes a pattern-based mixture of experts model for time series prediction. In this approach time series patches are first extracted and embedded into a sequence of "tokens". This sequence is then encoded with time and frequency encoders and mined for representative patterns that can aid prediction via a mixture of experts approach.

**Strengths:**

The paper is well written and relatively easy to follow. The proposed approach is interesting and explores time and frequency aspect of time series prediction. Authors conduct extensive experiments on real world datasets showing that the purposed approach outperforms a number of baselines.

**Weaknesses:**

I have several concerns about this method. First, it is quite complex, the loss in Eq 12 has three terms and both L_{PI_t} and L_{PI_f} have three tunable hyper parameters \alpha, \beta and \nu (Eq 5, 6 and 9). So there are at least six hyper parameters to tune in addition to balancing the three loss terms that likely requires more hyper parameters. There is nothing in the experimental section on these hyper parameters and no ablation study is conducted. I've worked with these datasets myself, they are quite small and easy to overfit to which raises a major concern when one has to tune 6+ hyper parameters just in the loss function. Second, the proposed mixture of experts in Eq 10 and 11 looks nearly identical to self attention with the exception of the top-k selection. So I suspect that this method will, despite significant added complexity, have a similar representation power to simply stacking attention layers over patches, as all operations including subspace bases affinity are nearly identical to appropriately parametrized self attention.

**Questions:**

Do you have ablation study on the hyper parameters in the loss function? Which ranges did you try and how sensitive are the results to the particular choice? Also, did you have to balance the contribution of the three terms in the loss?

---

> ### Author Response · Authors · 2024-11-20
> **Response to Reviewer JuoR**
>
> We thank the reviewer for offering the valuable feedback. Below we give a point-by-point response to your concerns and suggestions.
>
> > **Q1. The ablation study on the hyper parameters.**
>
> 1. In our work, we performed a grid search to optimize the hyperparameter **$\alpha_t$ = {0.0001, 0.001, 0.01}** and **$\alpha_f$ = {0.0001, 0.001, 0.01}** as shown in $\underline{\text{Figure 10 in Appendix P of the revised paper}}$, which were ultimately fixed at $10^{-3}$ in our experiments.
>
> 2. **$\eta$** is set to the same value as $d$ as the dimension of $\mathbf{D}^{(j)} \in \mathbf{R}^{q \times d}$ to control the smoothness.
>
> 3. Additionally, we conducted a grid search to optimize the balance factors **$\beta_t$ = {0.01, 0.05, 0.1, 0.5, 1}** and **$\beta_f$ = {0.01, 0.05, 0.1, 0.5, 1}**. The forecasting performance under different parameter values is displayed in $\underline{\text{Figure 10 in Appendix P of the revised paper}}$. Based on these results, we draw the following observations:
>
> * First, the performance is affected when the value of $\beta_t$ and $\beta_f$ is too low, indicating that the proposed **clustering objective is crucial for distinguishing patterns** effectively.
> * Second, excessively large values of $\beta_t$ and $\beta_f$ also degrade performance. One plausible explanation is that the excessive value **influences the learning of the inherent structure** of original data, which leads to disturbances of the embedding space.
> * Overall, we recommend setting $\beta_t$ and $\beta_f$ to around **0.1** for optimal performance.
>
> These analyses have also been included in $\underline{\text{Appendix P of revised paper}}$.
>
> 4. Finally, with the above hyperparameter tuning, the individual loss terms are already balanced, making it unnecessary to introduce additional balancing for the three terms in the final loss.
>
> 5. We employed **regularization techniques** such as early stopping to mitigate overfitting. These strategies, combined with **careful hyperparameter tuning**, help us maintain a balance between model complexity and predictive performance. Furthermore, our extensive experiments, as presented in the paper, demonstrate stable and competitive performance across various datasets, highlighting the **robustness** of our model.
>
> Through these efforts, we believe that our model is **not prone to overfitting**, and its generalization capabilities have been rigorously tested.
>
> > **Q2. Comparison with the stacking attention layers.**
>
> * The key distinction between TFPS and the stacking attention layers lies in their **modeling approaches**:
>
> The stacking attention layers adopts the **Uniform Distribution Modeling** strategy for all data (as presented in the $\underline{\text{Introduction of original submission}}$), whereas TFPS follows an **MoE structure** [1].
>
> Specifically, TFPS leverages the **Pattern Identifier** as a **gating network** to dynamically select relevant experts based on the input patterns. This distribution-aware approach allows each expert to specialize in different data patterns, thereby improving overall performance.
>
> [1] Shazeer N, Mirhoseini A, Maziarz K, et al. Outrageously Large Neural Networks: The Sparsely-Gated Mixture-of-Experts Layer. ICLR, 2016.
>
> * In addition, we have added **addition experiments** where we replaced the MoPE module with stacked self-attention layers, keeping all other components and configurations identical.
> The results demonstrate that our proposed method significantly outperforms the stacked attention layers, which validates the importance of the Top-K selection and pattern-aware design in enhancing the model's representation capacity. In contrast, self-attention typically treats all data points uniformly, which may limit its ability to capture subtle distribution shifts or evolving patterns across patches.
>
> These analyses have also been included in the $\underline {\text{Appendix R of revised paper}}$.
>
> | | ETTh1-96 | ETTh1-192 | ETTh1-336 | ETTh1-720 | ETTh2-96 | ETTh2-192 | ETTh2-336 | ETTh2-720 |
> | - | - | - | - | - | - | - | - | - |
> | **TFPS** | **0.398** | **0.423** | **0.484** | **0.488** | **0.313** | **0.405** | **0.392** | **0.410** |
> | Attention Layers | 0.399 | 0.452 | 0.492 | 0.508 | 0.334 | 0.407 | 0.409 | 0.451 |

---

> ### Author Response · Authors · 2024-11-25
> **Request of Reviewer's attention and feedback**
>
> Dear Reviewer,
>
> We would like to kindly remind you that it has been 4 days since we submitted our rebuttal. We hope our response has addressed your concerns and would appreciate any feedback you may have.
>
> In line with your suggestions, we have made several improvements and provided detailed clarifications in the following aspects:
>
> * Appendix P is added to analyze the **hyperparameter selection** experiments.
> * The **robustness** of TFPS is analyzed.
> * Appendix R elaborates on **the differences between TFPS and stacking attention layers**.
>
> We have also conducted additional experiments, visualizations, and ablations to support our findings, all of which are now included in the $\underline{\text{Revised Paper}}$.
>
> Thank you again for your thoughtful review. We look forward to your response and are happy to provide further clarification on any additional points.

---

> ### Author Response · Authors · 2024-12-02
>
> Dear Reviewer,
>
> We appreciate your efforts in reviewing our paper. We have:
>
> > **1. Clarified the hyperparameter selection strategy.**
>
> > **2. Clarified the difference between TFPS and stacking attention layers.**
>
> As the Author-Reviewer discussion period is nearing its conclusion, we would like to kindly inquire whether our most recent response has addressed your concerns. If so, we kindly hope that you can reconsider your rating of our work.
>
> Please feel free to reach out if there are any remaining points that need clarification.
>
> Best regards,
>
> The Authors

---

### Official Review · Reviewer_fcr7 · 2024-11-09

**Soundness:** 3
**Presentation:** 2
**Contribution:** 3
**Rating:** 6
**Confidence:** 4

**Summary:**

The paper addresses the challenge of distributional shift in time series data by proposing a novel framework called the Time Frequency Pattern-Specific (TFPS) architecture. This method is designed to effectively model complex patterns in time series, enhancing the model's ability to manage distributional shifts more robustly.

**Strengths:**

1. The paper is well written and easy to follow.
2. The paper is well motivated.
3. The method shows promising results compared to the baselines.

**Weaknesses:**

1. The number of subspace variables $k$, does not appear to be clearly specified in the paper. Given that these variables are essential for modeling the patterns of patches, it is crucial to understand how to set an appropriate value for $k$. The authors should provide further discussion on this aspect. Additionally, an ablation study examining the impact of different values of $k$ would be beneficial.

2. The paper doesn't specify how the subspace variables are initialized. I think the initialization of subspace variables is crucial to achieve optimal performance.

3. Since the subspace variables is utilized for clustering. It is better to provide the relevant visualization (e.g., tsne) to show the effectiveness.

4. Some important information, such as the number of MoPE, the value of hyper-parameters $\alpha$, $\beta$ used for the results in the Table 2 are not clearly specified.

5. What do the notations $K_t$ and $K_f$ mean? The paper doesn't provide the clear definition.

**Questions:**

see weakness

---

> ### Author Response · Authors · 2024-11-20
> **Response to Reviewer fcr7**
>
> We thank the reviewer for offering the valuable feedback. Below we give a point-by-point response to your concerns and suggestions.
>
> > **Q1. The number of subspace variables $k$.**
>
> The number of subspace variables **$k$ is equivalent to the number of experts $K$**. To improve clarity, we have unified the notation of $k$ in $\underline{\text{Section 3.5 line 288-293 of revised paper}}$.
>
> Additionally, we have conducted experiments to analyze the impact of different values of $K$, as detailed in $\underline{\text{Section 4.6 of original submission}}$, and the corresponding results are thoroughly discussed in the paper.
>
> > **Q2. The initialization of subspace.**
>
> In our method, the subspace variables $\mathbf{D} \in \mathbf{R}^{Kd \times Kd}$ are initialized with a **random Gaussian distribution** to ensure sufficient expressive power [1]. This initialization facilitates **efficient learning of feature representations** in the early stages of training, while **iterative refinement** during training enhances representation accuracy. For clarity, we have detailed this initialization process $\underline{\text{Appendix G Algorithm 1 of revised paper}}$.
>
> [1] Jiang Z, Zheng Y, Tan H, et al. Variational deep embedding: A generative approach to clustering. CoRR, 2016.
>
> > **Q3. The visualization of the clustering subspace.**
>
> We have included the **t-SNE** visualization of the learned embedded representation on the ETTh1 dataset in $\underline{\text{Appendix L of revised paper}}$.
> In Figure 9 (a), where the pattern identifier is replaced with a linear layer, the representation lacks clear clustering structures, resulting in **scattered and indistinct** groupings.
> In contrast, Figure 9 (b) demonstrates the representation learned by the proposed method, which effectively captures **discriminative** features and reveals significantly **clearer clustering patterns**.
>
> > **Q4. The value of MoPE, $\alpha$, $\beta$.**
>
> 1. In our work, we conducted a grid search to optimize the number of **MoPEs** in the time domain **$K_t$ = {1, 2, 4, 8}** and the number of experts in the frequency domain **$K_f$ = {1, 2, 4, 8}**, as detailed in $\underline{\text{Section 4.6 of original submission}}$.
>
> 2. Additionally, we performed a grid search to optimize the hyperparameter **$\alpha_t$ = {0.0001, 0.001, 0.01}** and **$\alpha_f$ = {0.0001, 0.001, 0.01}** as shown in $\underline{\text{Figure 10 in Appendix P of the revised paper}}$, which were ultimately fixed at $10^{-3}$ in our experiments.
>
> 3. Finally, we conducted a grid search to optimize the balance factors **$\beta_t$ = {0.01, 0.05, 0.1, 0.5, 1}** and **$\beta_f$ = {0.01, 0.05, 0.1, 0.5, 1}**. The forecasting performance under different parameter values is displayed in $\underline{\text{Figure 10 in Appendix P of the revised paper}}$. Based on these results, we draw the following observations:
>
> * First, the performance is affected when the value of $\beta_t$ and $\beta_f$ is too low, indicating that the proposed **clustering objective is crucial for distinguishing patterns** effectively.
> * Second, excessively large values of $\beta_t$ and $\beta_f$ also degrade performance. One plausible explanation is that the excessive value **influences the learning of the inherent structure** of original data, which leads to disturbances of the embedding space.
> * Overall, we recommend setting $\beta_t$ and $\beta_f$ to around **0.1** for optimal performance.
>
> These analyses have also been included in $\underline{\text{Appendix P of revised paper}}$.
>
> > **Q5. The explanation of $K_t$ and $K_f$.**
>
> $K_t$ and $K_f$ refer to the number of experts in the time-domain and frequency-domain components, respectively. We have clarified this definition in $\underline{\text{Section 4.1 of revised paper}}$ for better readability.

---

> ### Author Response · Authors · 2024-11-25
> **Request of Reviewer's attention and feedback**
>
> Dear Reviewer,
>
> We kindly remind you that it has been 4 days since we posted our rebuttal. Please let us know if our response has addressed your concerns.
>
> Following your suggestion, we have answered your concerns and improved the paper in the following aspects:
>
> * We clarify the notation of **subspace variables $K$, $K_t$, and $K_f$**, and explain the **subspace initialization operations**.
> * We have provided the implementation details for the **visualization subspace**.
> * Appendix P is added to analyze the **hyperparameter selection** experiments.
>
> We have provided extensive experiments, visualization, and ablations to support our insight. All of these results have been included in the $\underline{\text{Revised Paper}}$.
>
> Thanks again for your valuable review. We are looking forward to your reply and are happy to answer any future questions.

---

> > ### Comment · Reviewer_fcr7 · 2024-11-27
> > **Reply**
> >
> > Thank you for your response. It addresses my concerns. Based on the responses provided to me and other reviewers, I believe the paper has been improved. I will update my score accordingly.

---

> > > ### Author Response · Authors · 2024-11-28
> > > **Many thanks for your response and raising score**
> > >
> > > We sincerely thank the reviewer for replying. Meanwhile, we are deeply grateful for the reviewer's thoughtful feedback and for acknowledging our efforts. Thank you once again for your support and encouragement!

---

### Official Review · Reviewer_ENp4 · 2024-11-12

**Soundness:** 3
**Presentation:** 3
**Contribution:** 3
**Rating:** 6
**Confidence:** 4

**Summary:**

The paper innovatively addresses the diversity of time series patterns by introducing a Mixture of Experts (MoE) approach for decoupled modeling. It leverages a unique PI module as the gating function to route different patterns to the appropriate experts. The experimental results convincingly demonstrate the superiority of this method, complemented by comprehensive ablation studies validating the effectiveness of each component. The writing is clear, with the authors articulating the problem and solution effectively. However, the design of the expert models appears somewhat simplistic, and the decoupling of time and frequency domains is already a well-established approach in time series analysis, making it less of an innovative contribution. Overall, the method shows sufficient technical depth and originality, despite some areas that could benefit from further enhancement.

**Strengths:**

The authors conducted a detailed experimental analysis and identified that different time series exhibit unique patterns. Traditional models typically employ a unified modeling approach for these varying patterns, which can limit their effectiveness. To address this issue, the authors propose a Mixture of Experts (MoE) approach that enables tailored modeling for different patterns. Additionally, they decompose the time series data into two dimensions: time and frequency, using a dual encoder to model both types of information separately. To ensure that the MoE effectively learns distinct patterns, the authors introduce a novel gating mechanism. This mechanism leverages subspace affinity to allocate sequences with different patterns to the appropriate experts.

**Weaknesses:**

The decoupling of time-domain and frequency-domain information, as discussed in the paper, is already a common approach in the field of time series analysis and cannot be considered a novel contribution. Additionally, regarding the MoE (Mixture of Experts) section, the use of multiple MLPs seems more akin to employing multiple output predictors with weighted combinations rather than a true MoE structure. Lastly, with respect to the normalization technique, would the authors consider adding experimental results comparing their approach to REVIN [1]? REVIN has shown strong performance across various time series forecasting tasks and could provide a valuable benchmark.


[1] Kim T, Kim J, Tae Y, et al. Reversible instance normalization for accurate time-series forecasting against distribution shift[C]//International Conference on Learning Representations. 2021.

**Questions:**

1. In the PI module, the number of experts K is currently adjusted manually through parameter tuning. Is it possible to directly relate the value of K to the number of patterns inherent in the time series data itself?
2. On line 317, there appear to be two identical objectives listed, which seems to be a typographical error.

---

> ### Author Response · Authors · 2024-11-20
> **Response to Reviewer ENp4**
>
> We thank the reviewer for offering the valuable feedback. Below we give a point-by-point response to your concerns and suggestions.
>
> > **W1. The innovation of combining time and frequency domain.**
>
> Our contribution goes beyond simply decoupling time-domain and frequency-domain information **(Dual-Domain Encoder)**. The novelty of our method lies in identifying multiple patterns within the data **(Pattern Identifier)** and assigning them to specialized experts for modeling **(Mixture of Pattern Experts)**. This effectively addresses distribution shifts in time series data, which is a significant challenge in time series forecasting and leads to a significant improvement in forecasting performance.
>
> > **W2. An explanation of MoE section.**
>
> * The key distinction between TFPS and the weighted multi-output predictor lies in their **modeling approaches**.
>
> The weighted multi-output predictor adopts the **Uniform Distribution Modeling** strategy for all data (as presented in the $\underline{\text{Introduction of original submission}}$), whereas TFPS follows an **MoE structure** [1].
>
> Specifically, TFPS leverages the **Pattern Identifier** as a **gating network** to dynamically select relevant experts based on the input patterns. This distribution-aware approach allows each expert to specialize in different data patterns, thereby improving overall performance.
>
> [1] Shazeer N, Mirhoseini A, Maziarz K, et al. Outrageously Large Neural Networks: The Sparsely-Gated Mixture-of-Experts Layer. ICLR, 2016.
>
> * In addition, we have added **addition experiments** where we replaced the MoPE module with weighted multi-output predictor, keeping all other components and configurations identical.
> The results demonstrate that our proposed method significantly outperforms the multi-output predictor, which validates the importance of the Top-K selection and pattern-aware design in enhancing the model's representation capacity. In contrast, multi-output predictor typically treats all data points uniformly, which may limit its ability to capture subtle distribution shifts or evolving patterns across patches.
>
> These analyses have also been included in the $\underline {\text{Appendix R of revised paper}}$.
>
> | | ETTh1-96 | ETTh1-192 | ETTh1-336 | ETTh1-720 | ETTh2-96 | ETTh2-192 | ETTh2-336 | ETTh2-720 |
> | - | - | - | - | - | - | - | - | - |
> | **TFPS (Ours)** | **0.398** | **0.423** | **0.484** | **0.488** | **0.313** | **0.405** | **0.392** | **0.410** |
> | Multi-output Predictor | 0.403 | 0.435 | 0.492 | 0.491 | 0.317 | 0.407 | 0.399 | 0.425 |
>
> > **W3. Compared with RevIN.**
>
> We have conducted experiments comparing our approach to RevIN, as presented in $\underline{\text{Appendix E.2, Table 6 of original submission}}$. However, we sincerely apologize for the error in the average MSE statistics reported in Table 4. This issue has been corrected in $\underline{\text{Table 4 of the revised paper}}$.
>
> The updated average MSE results for all normalization-based comparison methods are provided below. By leveraging distribution-specific modeling, TFPS achieves significant improvements, with an average MSE reduction of 18.9%.
>
> | Dataset |  TFPS (Ours) | + SIN (2024) | + SAN (2023) | + Dist-TS (2023) | + NST  (2022)| + RevIN (2021)|
> | - | - | - | - | - | - | - |
> | ETTh1 | 0.448 | 0.458 | **0.447** | 0.461 | 0.456 | 0.463 |
> | ETTh2 | **0.380** | 0.501 | 0.403 | 1.005 | 0.481 | 0.465 |
> | ETTm1 | **0.395** | 0.409 | 0.377 | 0.422 | 0.411 | 0.415 |
> | ETTm2 | **0.276** | 0.437 | 0.287 | 0.759 | 0.315 | 0.310 |
> | Weather | **0.241** | 0.326 | 0.279 | 0.398 | 0.268 | 0.268 |
>
> > **Q1. Adaptively discovering time-series patterns.**
>
> We fully agree with your perspective on the selection of $K$. Directly linking $K$ to the number of inherent patterns in time series data is indeed a more practical and insightful approach. It would eliminate the need for manual tuning and allow for a more accurate representation of the data's intrinsic structure.
>
> However, due to the unknown nature of patterns in time series data, such an approach requires a mechanism to automatically detect the number of patterns. In a separate study, we have explored this idea using a **Dirichlet Process Gaussian Mixture Model** for **unsupervised pattern discovery**, followed by forecasting. While initial results are promising, this research is still in progress and requires further refinement.
>
> Your observation resonates strongly with our research direction, and we appreciate this insightful suggestion.
>
> > **Q2. Typographical error.**
>
> We apologize for the typographical error. We have corrected this in the revised paper.

---

> ### Author Response · Authors · 2024-11-25
> **Request of Reviewer's attention and feedback**
>
> Dear Reviewer,
>
> Thanks for your valuable and constructive review, which has inspired us to improve our paper further substantially. This is a kind reminder that it has been 4 days since we posted our rebuttal. Please let us know if our response has addressed your concerns.
>
> Following your suggestions, we have provided the following revisions to our paper:
>
> * **Further clarification of our innovative points**.
> * **Appendix R is added to elaborate on the differences between TFPS and weighted multi-output prediction** in terms of modeling strategy and experimental demonstration.
> * Further discussion on the prospect of **"adaptively discovering time series patterns."**
>
> In this paper, we propose the TFPS as a simple but effective model and provide extensive experiments, visualization, and ablations to support our insight. All the revisions are included in the $\underline{\text{Revised Paper}}$.
>
> Sincere thanks for your dedication! We are looking forward to your reply.

---

> > ### Comment · Reviewer_ENp4 · 2024-11-26
> > **Thanks for the rebuttal.**
> >
> > Thanks. I appreciate your efforts in rebuttal. I will keep my score.

---

> > > ### Author Response · Authors · 2024-11-27
> > > **Many thanks for your response**
> > >
> > > We sincerely thank the reviewer for replying. Meanwhile, we are deeply grateful for the reviewer's thoughtful feedback and for acknowledging our efforts. Thank you once again for your support and encouragement!

---

### Author Response · Authors · 2024-11-20
**Summary of Revisions**

We sincerely thank all the reviewers for their insightful reviews and valuable comments, which are instructive for us to improve our paper further.

This paper introduces TFPS, a robust and innovative model for time series forecasting, which tackles distribution shifts through a novel time-frequency pattern-specific perspective. By leveraging the complementary strengths of time-domain and frequency-domain features, combined with adaptive pattern-specific experts, TFPS effectively captures diverse temporal patterns and achieves superior forecasting performance.
**Experimentally, TFPS obtains significant performance in 9 well-established benchmarks with competitive run-time efficiency for distribution shift in long-term forecasting tasks.**

The reviewers generally held positive opinions of our paper, noting that the proposed method "**shows sufficient technical depth and originality**"; the method is "**interesting**" with "**well motivation**" and "**promising results**"; this paper is "**well-written**", and "**easy to follow**"; we have provided "**extensive experiments**" which "**convincingly demonstrate the superiority**", as well as "**comprehensive ablation studies**".

The reviewers also raised insightful and constructive concerns. We made every effort to address all the concerns by providing detailed clarification and requested results. Here is the summary of the major revisions:

- **Elaborate differences among TFPS and multiple predictors, stack self-attention layers (Reviewer ENp4, JuoR)**: We clarify that TFPS fundamentally differs from these approaches. TFPS employs a pattern identifier that dynamically selects the most relevant expert for modeling distinct patterns, but other approaches only adopt a uniform distribution modeling strategy.

- **Provide visualization of embedded representation (Reviewer fcr7)**: Following the reviewer's suggestion, we add the relevant visualization of embedding representation where TFPS can distinguish between different patterns precisely.

- **Provide the hyperparameters sensitivity experiments (Reviewer fcr7, JuoR, n6LR)**: As per the reviewer's request, we clarify the hyperparameter search strategy employed in this paper, along with the recommended selection range and criteria for optimal selection.

- **Provide the performance under treating seq\_len as a hyperparameter (Reviewer n6LR)**: Following the reviewer's suggestion, we provide the experimental results comparing seq\_len as a hyperparameter. TFPS outperforms other baselines significantly under unified hyperparameter and hyperparameter searching.

- **Add comparison with Koopa, SOLID, OneNet, MoLE, MoU, KAN4TSF (Reviewer LQda)**: Following the reviewer's suggestion, we have compared the above six baselines. Comparing to these new baselines, TFPS still performs best.

**After 7 full days of experiments, we have newly added more than 370 new experiment results, and 9 appendices to address the mentioned issues. All the revisions have been included in the $\underline {\text{revised paper highilghted in purple}}$.**

The valuable suggestions from reviewers are very helpful for us to revise the paper to a better shape. We'd be very happy to answer any further questions.

Looking forward to the reviewer's feedback.

---

### Meta-Review · Area_Chair_DfoA · 2024-12-22

**Metareview:**

This paper presents a pattern-based mixture of experts model for time series prediction. Reviewers agreed that this paper is well written and easy to follow, and the paper aims to address a critical problem, i.e., patch-level distributional shift in time series forecasting. Meanwhile, reviewers raised many concerns about novelty, technical details, comparisons with baselines, missing related work, etc. During the post-rebuttal discussions, reviewers still believed that some major issues still remain. For instance, the paper fails to exhibit a significant efficiency advantage over SOTA baselines. Overall, this work in its current version is not ready for publication at ICLR.

**Additional Comments On Reviewer Discussion:**

Reviewers had many concerns about novelty, technical details, comparisons with baselines, missing related work, etc. The authors provided detailed responses, but they cannot fully address the reviewers' concerns, such as the comparisons with SOTA methods and the efficiency of the proposed method.

---

### Decision · Program_Chairs · 2025-01-22

Reject